# ALIGN-THEN-STEER: ADAPTING THE VISION-LANGUAGE-ACTION MODELS THROUGH UNIFIED LATENT GUIDANCE

**Yang Zhang**[1,2*], **Chenwei Wang**[1,3*], **Ouyang Lu**[1,4*], **Yuan Zhao**[1], **Yunfei Ge**[1],
**Zhenglong Sun**[3], **Xiu Li**[2], **Chi Zhang**[1], **Chenjia Bai**[1†], **Xuelong Li**[1†]
[1]Institute of Artificial Intelligence, China Telecom, [2]Tsinghua University,
[3]The Chinese University of Hong Kong, Shenzhen, [4]Northwestern Polytechnical University
`baicj@chinatelecom.cn`, `xuelong_li@ieee.org`

## ABSTRACT

Vision-Language-Action models (VLAs) pre-trained on large, diverse datasets show remarkable potential for general-purpose robotic manipulation. However, a primary bottleneck remains in adapting these models to downstream tasks, especially when the robot's embodiment or the task itself differs from the pre-training data. This discrepancy leads to a significant mismatch in action distributions, demanding extensive data and compute for effective fine-tuning. To address this challenge, we introduce **Align-Then-stEer (`ATE`)**, a novel, data-efficient, and plug-and-play adaptation framework. `ATE` first aligns disparate action spaces by constructing a unified latent space, where a variational autoencoder constrained by reverse KL divergence embeds adaptation actions into modes of the pre-training action latent distribution. Subsequently, it steers the diffusion- or flow-based VLA's generation process during fine-tuning via a guidance mechanism that pushes the model's output distribution towards the target domain. We conduct extensive experiments on cross-embodiment and cross-task manipulation in both simulation and real world. Compared to direct fine-tuning of representative VLAs, our method improves the average multi-task success rate by up to **9.8%** in simulation and achieves a striking **32% success rate gain** in a real-world cross-embodiment setting. Our work presents a general and lightweight solution that greatly enhances the practicality of deploying VLA models to new robotic platforms and tasks.

## 1 INTRODUCTION

Vision-Language-Action models (VLAs) (Brohan et al., 2022; Zitkovich et al., 2023; Kim et al., 2024; Black et al., 2024) have emerged as a powerful solution to empower general-purpose robotic manipulation with broad generalization in tasks and environments. Inspired by the success in scaling data and parameters of Large Language Models (LLMs) and Visual-Language Models (VLMs), the VLA models extend the pre-trained VLM models (Li et al., 2023; Driess et al., 2023; Beyer et al., 2024; Wang et al., 2024) by integrating an action expert module to generate low-level actions, thereby inheriting the common knowledge, semantic reasoning, and instruction-following capabilities of VLMs to facilitate policy learning. Specifically, a representative training recipe for VLAs typically consists of two phases: a (i) *pre-training phase*: pretrain the entire VLA model that contains a strong VLM and an action expert on a large corpus containing cross-embodiment datasets (Jang et al., 2021; Collaboration et al., 2023; Khazatsky et al., 2024), which spans diverse robots and tasks to learn a general visuomotor prior at scale, and an (ii) *adaptation phase*: fine-tune the VLA model on a narrowly curated dataset tailored to a particular robot platform and associated tasks, aiming to enhance the manipulation abilities in specific downstream domains.

However, the two-phase paradigm for pre-trained VLAs faces a significant challenge when adapting to downstream domains with different embodiments. For example, pre-training datasets often

---

[*]Equal contribution
[†]Corresponding authors

use single-arm robots, while target embodiments might be dual-arm or humanoid. Since VLAs are tied to imitation learning and its action labels, the resulting action space and distribution mismatch between pre-training and adaptation phases create a substantial bottleneck. Even with the same robot, differences in tasks and setup lead to similar distribution discrepancies, often requiring extensive data collection to resolve. While recent approaches focus on reducing training costs at the parameter level (Kim et al., 2024; Zhang et al., 2025b) or with different architectural designs (Kim et al., 2025; Pertsch et al., 2025), they often struggle to address discrepancies in task distributions and overlook the embodiment mismatches that are significantly more challenging to resolve. This motivates our central question: how can we achieve *VLA adaptation for both cross-embodiment and cross-task scenarios under a limited data regime?*

To this end, we propose a data-efficient adaptation method, named **Align-Then-stEer (`ATE`)**, for cross-embodiment and cross-task VLA adaptation by explicitly mitigating action-distribution mismatch via a two-stage strategy. In the first stage, we construct a unified latent action space to bridge the domain gap between embodiments and tasks involved during pre-training and the adaptation phases. It is achieved by training a Variational Autoencoder (VAE) on the pre-training action-labeled dataset to obtain a latent prior, followed by learning a smaller VAE on limited adaptation data with a reverse KL divergence constraint toward the learned pre-trained latent distribution. Thanks to the mode-seeking behavior of reverse KL divergence, this encourages the encoded latent of the target embodiment's actions to be embedded into a specific mode of the learned prior, yielding a unified and hierarchically structured latent space. In the second stage, inspired by classifier-guided diffusion models (Dhariwal & Nichol, 2021; Bansal et al., 2023), we design a guidance function grounded in the shared latent space that can be seamlessly integrated into the training objective of diffusion- or flow-based VLAs. This classifier guidance steers the output distribution of the pre-trained VLA toward that of the target embodiment and task, enabling faster and more precise adaptation under limited data. Notably, our method requires no modification to the original VLA architecture and introduces only negligible overhead, i.e., adaptation is highly efficiently achieved by training two separate lightweight VAEs. An overview of our method is given in Fig. 1. Our key contributions are summarized as follows:

- We propose a novel alignment strategy to bridge the action-distribution gap between pre-training and adaptation. By leveraging the mode-seeking behavior of reverse KL, our method constructs a unified, structured latent space that embeds target actions into modes of the pre-training latent distribution.
- We introduce a classifier-guidance mechanism that operates within this unified latent space to explicitly steer the VLA's output distribution. This enables precise and rapid adaptation to new embodiments and tasks, without requiring additional data.
- The proposed `ATE` framework is model-agnostic and plug-and-play, seamlessly integrating with various diffusion- or flow-based VLAs without modifying their architecture and with negligible computational overhead.
- We validate our method across different embodiments and tasks in both simulated and real-world manipulation settings. The result demonstrates consistent improvements over baselines that rely on direct supervised fine-tuning, highlighting our approach's superior adaptation efficiency.

## 2 RELATED WORKS

**Policy Adaptation of VLAs.** Despite the promise of VLAs for general-purpose robot manipulation, current approaches still suffer from high fine-tuning costs for embodiment and task mismatches. Several methods address this by constructing shared latent action models (Ye et al., 2025; Bu et al., 2025; NVIDIA, 2025), performing kinematics retargeting (Bauer et al., 2025; Yuan et al., 2025), or employing lighter-weight fine-tuning techniques like LoRA-based updates (Wen et al., 2025b) and dynamic layer activation (Zhang et al., 2025a). While these approaches improve training efficiency or establish some form of shared space, they are often limited by architectural constraints (e.g., auto-regressive models) or fail to effectively bridge the significant action distribution gap between domains. In contrast, we propose a novel perspective that we can use the mode-seeking property of VAEs to explicitly bridge the distribution gap between pre-training and adaptation data. This is complemented by a latent guidance mechanism that is compatible with both diffusion- or flow-based VLAs. More discussions on VLAs and constructing unified action spaces is deferred to Appendix A.1.

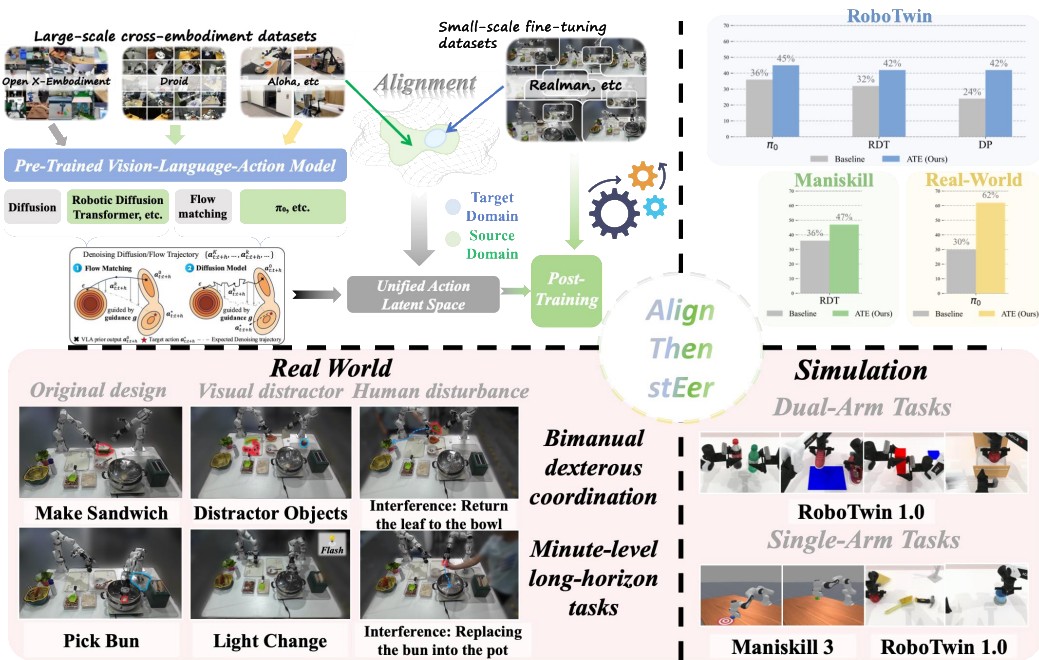

Figure 1: We present ATE, a plug-and-play adaptation framework for pre-trained VLAs. By aligning disparate action spaces into a unified latent representation and steering generation via guidance, ATE enables data-efficient cross-task and cross-embodiment adaptation. Evaluations on RoboTwin, ManiSkill, and a real-world dual-arm RealMan 7-DoF robot show substantial gains in success rates.

## 3 PRELIMINARIES

**Training objective of VLAs.** The training of VLA models typically contains two stages: (i) obtain an initial policy via pre-training the model on a cross-embodiment action-labeled dataset $\mathcal{D}_{\text{pretrain}}$, where we denote the set of embodiments included in it as $\mathcal{M}_{\text{pretrain}} = \{\mathcal{E}_1, \mathcal{E}_2, \ldots, \mathcal{E}_n\}$; (ii) finetune the policy on a narrow but high-quality dataset $\mathcal{D}_{\text{adaptation}}$ collected from the target embodiment $\mathcal{E}_{\text{target}}$ for downstream applications. Formally, the optimization objective of the VLA policy $\pi_\theta$ is to maximize the log-likelihood of an action chunk $\mathbf{a}_{t:t+h}$ given an observation $\mathbf{o}_t$ and a language-format task instruction $l$ sampled from the dataset, which can be formulated as $\max_\theta \mathbb{E}_{(\mathbf{a}_{t:t+h}, \mathbf{o}_t, l) \sim \mathcal{D}} [\log \pi_\theta(\mathbf{a}_{t:t+h} | \mathbf{o}_t, l)]$, where $\mathbf{o}_t$ contain single-view or multi-view RGB observations $\mathbf{I}_t^1, \mathbf{I}_t^2, \ldots, \mathbf{I}_t^n$, and the proprioceptive state $\mathbf{q}_t$ usually describes the joint position and angle of the robot.

**Diffusion Models and Flow Matching.** Diffusion models (Sohl-Dickstein et al., 2015; Ho et al., 2020; Song et al., 2021) are generative models where data generation is modeled as a denoising process. Consider a forward diffusion process that gradually adds Gaussian noise to samples $\mathbf{x}_0 \sim q(\mathbf{x}_0)$ over $T$ steps and produces a sequence of noisy samples $\mathbf{x}_1, \ldots, \mathbf{x}_T$ that satisfies $q(\mathbf{x}_{1:T} | \mathbf{x}_0) = \prod_{t=1}^T q(\mathbf{x}_i | \mathbf{x}_{i-1})$ and $q(\mathbf{x}_t | \mathbf{x}_{t-1}) = \mathcal{N}(\mathbf{x}_t; \sqrt{\alpha_t} \mathbf{x}_{t-1}, (1 - \alpha_t)I)$, where $\{\alpha_t \in (0,1)\}_{t=1}^T$ is a predefined noise scheduler. The denoising diffusion model reverses the above process to approximate the data distribution $p_\theta(\mathbf{x}_0)$ which is parameterized as $p_\theta(\mathbf{x}_0) = \int p(\mathbf{x}_{0:T}) d\mathbf{x}_{1:T} = \int p(\mathbf{x}_T) \prod_{t=1}^T p_\theta(\mathbf{x}_{t-1} | \mathbf{x}_t) d\mathbf{x}_{1:T}$. By minimizing the following denoising loss function introduced by Ho et al. (2020),

$$\mathcal{L}(\theta) = \mathbb{E}_{t, \mathbf{x}_0, \epsilon}[\|\epsilon - \epsilon_\theta(\sqrt{\bar{\alpha}_t} \mathbf{x}_0 + \sqrt{1 - \bar{\alpha}_t} \epsilon, t)\|^2], \text{ where } \bar{\alpha}_t = \prod_{i=1}^t \alpha_i, \qquad (1)$$

we can get a noise prediction network $\epsilon_\theta(\mathbf{x}_t, t)$ to perform the reverse diffusion process - it begins by sampling $\mathbf{x}_T \sim \mathcal{N}(0, I)$, and then iteratively applies the update

$$\mathbf{x}_{k-1} = \frac{1}{\sqrt{\alpha_k}} \left( \mathbf{x}_k - \sqrt{1 - \bar{\alpha}_k} \epsilon_\theta(\mathbf{x}_k, k) \right) + \sigma_k \epsilon_k, \qquad (2)$$

where $\epsilon_k \sim \mathcal{N}(0, I)$, and $\sigma_k$ is a function of $k$ and depends on the noise scheduler.

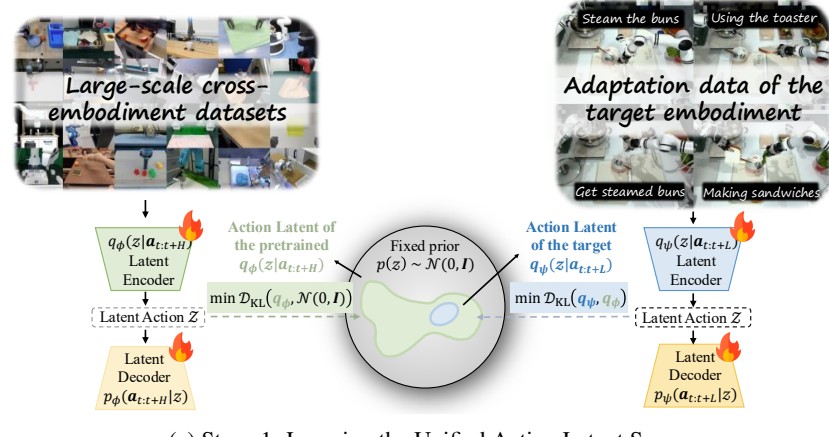

(a) Stage 1: Learning the Unified Action Latent Space

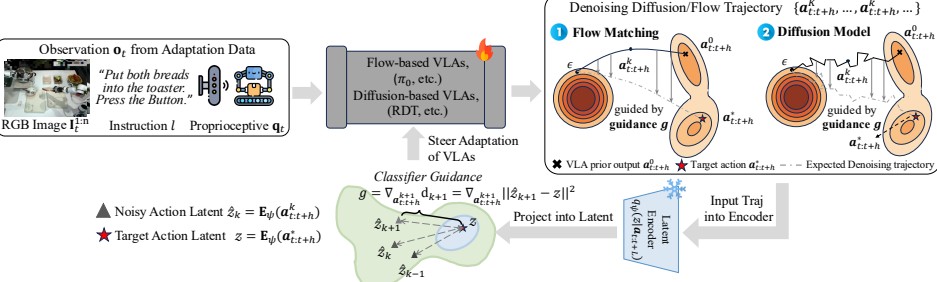

(b) Stage 2: Steering Efficient Adaptation with Latent Guidance

Figure 2: **The overview of ATE framework.** (a) In the first stage, we construct a unified action space to bridge the embodiment gap in pretraining and adaptation stages by utilizing the mode-seeking behavior of asymmetric VAEs. (b) In the second stage, we integrate classifier guidance in diffusion and flow-based VLAs to steer the pretrained policy towards the target action distribution with specific robot platforms.

Beyond denoising diffusion, flow matching is another generative modeling framework that implements data generation as a probability path from a known source distribution to the data target distribution (Lipman et al., 2023; Liu et al., 2023b; Lipman et al., 2024). Assuming the source sample and target sample are denoted as $\mathbf{x}_1$ and $\mathbf{x}_0$ respectively, and flow matching timesteps are denoted as $\tau \in [0, 1]$. By instantiating the path as a linear-Gaussian path given by $\mathbf{x}_\tau \sim \mathcal{N}(\tau \mathbf{x}_0, (1 - \tau)I)$, flow matching operates by first sampling noise from a standard Gaussian and then processes this noise through a deterministic process to produce a sample from the target distribution, similar to the denoising process. The flow $v_\theta$ that maintains the probability path is typically trained via optimizing a simple conditional flow matching loss function,

$$\mathcal{L}(\theta) = \mathbb{E}_{\tau, \mathbf{x}_0, \epsilon} \left[ \left\| v_\theta(\tau \mathbf{x}_0 + (1 - \tau)\epsilon, \tau) - (\mathbf{x}_0 - \epsilon) \right\|^2 \right]. \tag{3}$$

In robotic manipulation, there can be dozens of possible actions $\mathbf{a}_t$ to accomplish the task given the same language instruction $l$ and observation. Motivated by the capability of diffusion models and flow matching on providing high-precision continuous representation and modeling multi-modal distribution, recent VLAs (Black et al., 2024; Liu et al., 2024; Intelligence et al., 2025; Bjorck et al., 2025) have incorporated diffusion-based or flow-based action prediction to boost performance.

**Problem Setting.** In this work, we assume access to a pre-trained diffusion-based or flow-based policy, which can be either a VLA model or Vision-Action model like Diffusion Policy (Chi et al., 2023). Our goal is to adapt the pretrained model in $\mathcal{D}_{\text{pretrain}}$ to a specific embodiment $\mathcal{E}_{\text{target}}$ given an adaptation dataset $\mathcal{D}_{\text{adaptation}}$, where $\mathcal{E}_{\text{target}}$ is not be involved in pre-training, i.e., $\mathcal{E}_{\text{target}} \notin \mathcal{M}_{\text{pretrain}}$. In addition, the tasks contain in $\mathcal{D}_{\text{adaptation}}$ can also be significantly different from that in pre-training. For clarity, we use the superscript $k$ to denote the diffusion or flow-matching timestep, and the subscript $t$ to denote the trajectory timestep.

## 4 METHODOLOGY

The proposed $\texttt{ATE}$ algorithm mainly consists of two stages: (i) we extract a unified action latent space to compactly encompass the actions of different embodiments, thus bridging the domain gap between the embodiments in the pre-training and adaptation phase (details in Section 4.1); (ii) then, we design a guidance function grounded in the unified latent space and leverage the resulting classifier guidance to explicitly steer the fine-tuning towards our desired data distribution during adaptation (details in Section 4.2). Fig. 2 provides an overview of our entire procedure.

### 4.1 ALIGNMENT THROUGH UNIFIED ACTION LATENT SPACE

A pre-trained VLA acquires a general visuomotor prior by modeling distribution $p(\bar{\mathbf{a}}_{t:t+H-1}|\mathbf{o}_t, l)$, where action chunk $\bar{\mathbf{a}}_{t:t+H-1}$ of length $H$, observation $\mathbf{o}_t$ and language instruction $l$ are sampled from the pre-training data $\mathcal{D}_{\text{pretrain}}$. Such a prior effectively captures the action distribution $p_{\mathcal{D}_{\text{pretrain}}}(\bar{\mathbf{a}}_{t:t+H-1})$ from the pre-training embodiments $\mathcal{M}_{\text{pretrain}}$. However, due to the inherent differences in degrees of freedom (DoFs) and physical representation (e.g., joint angles, end-effector poses, and joint torques), the raw action-labeled data from the target embodiment $\mathcal{E}_{\text{target}}$ can vary significantly from the action generated by the pre-trained VLA. This leads to a substantial **domain gap** that hinders efficient adaptation. To bridge this gap, we employ two distinct Variational Autoencoders (VAEs) to encode the fundamentally disparate pre-training and adaptation action data into a single, unified, and compact action latent space.

**Learning Pre-training Action Latent.** We build a pre-training action-VAE $\mathcal{V}_{\text{pretrain}} := \{\mathbf{E}_\phi, \mathbf{D}_\phi\}$ to compress all possible action chunks in the pre-training data into a low-dimensional latent space $\mathcal{Z}$, which consists of a Transformer encoder $\mathbf{E}_\phi$ and a Transformer decoder $\mathbf{D}_\phi$. The encoder takes an action chunk $\bar{\mathbf{a}}_{t:t+H-1}$ of length $H$ as input and outputs the parameters of a Gaussian distribution $q_\phi(z|\bar{\mathbf{a}}_{t:t+H-1})$ to represent the action latent. Here, we instantiate $\mathcal{V}_{\text{pretrain}}$ as an InfoVAE (Zhao et al., 2017) rather than a vanilla VAE (Kingma & Welling, 2013). Formally, it is optimized by maximizing the log-likelihood term together with a KL regularization term towards the fixed latent prior distribution and a mutual information maximization term that encourages high mutual information between $\bar{\mathbf{a}}_{t:t+H-1}$ and $z$. The objective can be simplified as,

$$\mathcal{L}(\phi; \mathcal{D} \leftarrow \mathcal{D}_{\text{pretrain}}) = \mathbb{E}_{\mathcal{D}}\mathbb{E}_{q_\phi(z|\bar{\mathbf{a}}_{t:t+H-1})}[\log p_\phi(\bar{\mathbf{a}}_{t:t+H-1}|z)] -$$
$$(1-\alpha)\mathbb{E}_{\mathcal{D}}[D_{\text{KL}}(q_\phi(z|\bar{\mathbf{a}}_{t:t+H-1})\|p(z))] - (\alpha - \lambda - 1)D_{\text{KL}}(q_\phi(z)\|p(z)), \quad (4)$$

where $\mathcal{V}_{\text{pretrain}}$ is parameterized by $\phi$, and $\alpha$ and $\lambda$ are hyperparameters that control penalty strength. The latent prior $p(z)$ is set as a unit Gaussian $\mathcal{N}(0, I)$.

**Embedding Adaptation Action Latent into Pre-training Action Latent Space.** With the pre-training action VAE $\mathcal{V}_{\text{pretrain}}$, we can obtain the learned action latent distribution $q_\phi(z)$ by marginalizing over all action chunks in $\mathcal{D}_{\text{pretrain}}$. To create a unified action latent space, we train a separate adaptation action VAE $\mathcal{V}_{\text{adaptation}} := \{\mathbf{E}_\psi, \mathbf{D}_\psi\}$ which is parameterized as $\psi$, with a KL regularization towards the learned pre-training action latent distribution. Denoting the action chunk of length $L$ in the adaptation data as $\tilde{\mathbf{a}}_{t:t+L-1}$, the learning objective of $\mathcal{V}_{\text{adaptation}}$ is written as follows,

$$\mathcal{L}(\psi; \mathcal{D} \leftarrow \mathcal{D}_{\text{adaptation}}) = \mathbb{E}_{\mathcal{D}}\mathbb{E}_{q_\psi(z|\tilde{\mathbf{a}}_{t:t+L-1})}[\log p_\psi(\tilde{\mathbf{a}}_{t:t+L-1}|z)] -$$
$$(1-\alpha)\mathbb{E}_{\mathcal{D}}[D_{\text{KL}}(q_\psi(z|\tilde{\mathbf{a}}_{t:t+L-1})\|q_\phi(z))] - (\alpha - \lambda - 1)D_{\text{KL}}(q_\psi(z)\|q_\phi(z)). \quad (5)$$

The adaptation action VAE has the same architecture and variant as the pre-training VAE. Minimizing the reverse KL divergence $D_{\text{KL}}(q_\psi(z|\tilde{\mathbf{a}}_{t:t+L-1})\|q_\phi(z))$ instead of $D_{\text{KL}}(q_\phi\|q_\psi)$ ensures that the learned adaptation action latent distribution is mode-seeking (Bishop & Nasrabadi, 2006). Therefore, this effectively embeds the adaptation latent distribution $q_\psi(z)$ into one mode of the pre-trained action latent distribution $q_\phi(z)$, yielding a unified and compact latent space $\mathcal{Z}$.

**Implementation Details.** The purpose of using two VAEs is to bridge the distribution gap of the output action chunks between the pre-trained VLA and the adaptation data. Thus, the pre-training VAE $\mathcal{V}_{\text{pretrain}}$ uses the same action chunk length $H$ as in the pre-training phase, while the adaptation action VAE $\mathcal{V}_{\text{adaptation}}$ adopts the action chunk length $L$ used in the adaptation phase. Following the guidelines of Zhao et al. (2017), we replace the last KL term in both Eq. (4) and Eq. (5) with Maximum-Mean Discrepancy (Li et al., 2015; Dziugaite et al., 2015) for more tractable optimization. The learned pre-training action latent distribution $q_\phi(z)$ in Eq. (5) is approximated as $\mathcal{N}(\mu_\phi, \Sigma_\phi)$, where $\mu_\phi$ and $\Sigma_\phi$ are calculated over all sampled pre-training action latents. A detailed algorithm for training both VAEs is provided in Alg. 1. More details on the design of the Transformer-based VAE are also provided in Appendix C.

## 4.2 Classifier Guidance for Steering Adaptation

To facilitate efficient adaptation of a pre-trained diffusion or flow-based VLA, we introduce classifier guidance (Dhariwal & Nichol, 2021) to explicitly steer the fine-tuning process. The core idea is to design a guidance function that measures the discrepancy between the generated actions and the target action distribution within the unified latent space $\mathcal{Z}$, and then use the gradient of this function to guide the policy update. This approach enables highly efficient adaptation without requiring additional data budget. In the following, we first introduce the formulation of classifier guidance in diffusion and flow-based VLAs; then we detail the guidance function according to the learned latent action space.

**Classifier Guidance in Diffusion-based VLAs.** Note that the action latent $z$ is only determined by the action chunk $\mathbf{a}_{t:t+h}$, i.e., independent of the observation $\mathbf{o}_t$ and language instruction $l$. As the conditioning label we use for guidance is based solely on the latent $z$, we thus omit these conditions and treat it as an unconditional diffusion model $\pi_\theta(\cdot) := p_\theta(\cdot)$ for notational simplicity. A diffusion model typically predicts the noise $\epsilon_\theta(\mathbf{a}_{t:t+h}^k, k)$ that corrupts an initial action chunk $\mathbf{a}_{t:t+h}^0$ to a noisy version $\mathbf{a}_{t:t+h}^k$. To generate a desired action chunk conditioned on a label $y$, we have to sample from the conditional distribution $p_\theta(\mathbf{a}_{t:t+h}^k|y)$. Fortunately, as proven by Luo (2022), by Tweedie's Formula and Bayes' rule, we can obtain a calibrated noise prediction $\hat{\epsilon}$ that incorporates the guidance gradient $g = \nabla_{\mathbf{a}_{t:t+h}^k} \log p(y|\mathbf{a}_{t:t+h}^k)$, which corresponds to the score of the joint distribution of the action chunk and the given conditioning label $y$, as

$$\hat{\epsilon}(\mathbf{a}_{t:t+h}^k, k) := \epsilon_\theta(\mathbf{a}_{t:t+h}^k, k) - \sqrt{1 - \bar{\alpha}_t} \nabla_{\mathbf{a}_{t:t+h}^k} \log p(y|\mathbf{a}_{t:t+h}^k) = \epsilon_\theta(\mathbf{a}_{t:t+h}^k, k) - \sqrt{1 - \bar{\alpha}_t} g, \quad (6)$$

where we delay the discussion of the exact expression of $g$ subsequently. By incorporating this modified noise prediction into the original denoising objective, we explicitly steer the fine-tuning stage of the VLA. The final objective is reformulated as,

$$\mathcal{L}(\theta) = \mathbb{E}_{k,\epsilon,(\mathbf{o}_t,\mathbf{a}_{t:t+h}^0,l)\sim\mathcal{D}_{\text{adaptation}}} \left[ \left\| \epsilon - \epsilon_\theta(\mathbf{a}_{t:t+h}^k; k, \mathbf{o}_t, l) + \sqrt{1 - \bar{\alpha}_k} \cdot \lambda \cdot g \right\|^2 \right], \quad (7)$$

where $\lambda$ is a guidance scale that modulates the influence of the guidance.

**Classifier Guidance in Flow-based VLAs.** Similarly, for those VLAs that model the generation of action chunks with Flow Matching framework, we also provide an updated objective that directly incorporates the classifier guidance into the learning of the flow. As proven by Lipman et al. (2024), supposing a velocity field $v_\theta$ is an instance of linear Gaussian flows $\mathbf{x}^\tau = \tau \mathbf{x}^1 + (1 - \tau)\mathbf{x}^0$ where $q(\mathbf{x}^1)$ is the target data distribution and $p(\mathbf{x}^0) = \mathcal{N}(0, I)$ is the source distribution, the transformation between marginal velocity fields and the score function for unconditional distributions satisfies

$$v_\theta(\mathbf{x}^\tau) = \frac{1}{\tau} \mathbf{x}^\tau + \frac{1 - \tau}{\tau} \nabla_{\mathbf{x}^\tau} \log p_\theta(\mathbf{x}^\tau). \quad (8)$$

We refer readers to Lipman et al. (2024) for a detailed proof. Similarly, when we condition the velocity field $v_\theta(\mathbf{a}_{t:t+h}^\tau|y)$ on a label $y$, we have

$$\hat{v}_\theta(\mathbf{a}_{t:t+h}^\tau|y) = v_\theta(\mathbf{a}_{t:t+h}^\tau) + \frac{1 - \tau}{\tau} \nabla_{\mathbf{a}_{t:t+h}^\tau} \log p(y|\mathbf{a}_{t:t+h}^\tau). \quad (9)$$

The final objective for fine-tuning the pre-trained velocity field $v_\theta$ is thus rewritten as,

$$\mathcal{L}(\theta) = \mathbb{E}_{\tau,\epsilon,(\mathbf{o}_t,\mathbf{a}_{t:t+h}^0,l)\sim\mathcal{D}_{\text{adaptation}}} \left[ \left\| v_\theta\left(\mathbf{a}_{t:t+h}^\tau; \tau, \mathbf{o}_t, l\right) + \frac{1 - \tau}{\tau} \cdot \lambda \cdot g - (\mathbf{a}_{t:t+h}^0 - \epsilon) \right\|^2 \right]. \quad (10)$$

**Designing a Guidance Function for Efficient Adaptation.** The goal of fine-tuning a pre-trained VLA is to fit the action distribution of the adaptation data. Despite the inherent heterogeneity of actions from different embodiments, we can effectively measure a valid discrepancy between them by leveraging the unified and compact action latent space $\mathcal{Z}$, established in Section 4.1. To facilitate efficient adaptation, we design a guidance function that measures how closely an intermediate action chunk during the denoising process, $\hat{\mathbf{a}}_{t:t+h}^k$, aligns with the ground truth action chunk, $\mathbf{a}_{t:t+h}^0$, from the adaptation data. Inspired by Carvalho et al. (2023); Liang et al. (2025), we formulate the classifier induced by such a guidance function as an energy-based model,

$$p_\psi(y|\hat{\mathbf{a}}_{t:t+h}^k) = \frac{1}{Z_\psi} \exp(-\|\mathbf{E}_\psi(\hat{\mathbf{a}}_{t:t+h}^k) - \mathbf{E}_\psi(\mathbf{a}_{t:t+h}^0)\|^2),$$

where $\mathbf{E}_\psi$ is the encoder of the adaptation action VAE and $Z_\psi$ is a normalizing constant dependent on $\psi$. Then the classifier guidance $g$ can be easily calculated via,

$$g = \nabla_{\hat{\mathbf{a}}_{t:t+h}^k} \log p_\psi(y|\hat{\mathbf{a}}_{t:t+h}^k) \propto -\nabla_{\hat{\mathbf{a}}_{t:t+h}^k} \|\mathbf{E}_\psi(\hat{\mathbf{a}}_{t:t+h}^k) - \mathbf{E}_\psi(\mathbf{a}_{t:t+h}^0)\|^2, \qquad (11)$$

where we use $\mathbf{E}_\psi$ to encode both noisy actions and clean actions. One may ask whether the encoder $\mathbf{E}_\psi$ can robustly encode noisy action chunks. As identified by Zhao et al. (2017), the vanilla VAE (Kingma & Welling, 2013) suffers from issues such as posterior collapse and inaccurate amortized latent inference, while the InfoVAE effectively addresses these limitations, making it an appropriate alternative for our deliberately designed guidance function. Since the guidance relies on the ground truth action chunk, which is not available during VLA inference, we incorporate this guidance explicitly into the training objective, as shown in Eqs. (7) and (10). During training, we use the noisy action chunk $\mathbf{a}_{t:t+h}^k$ corrupted from its ground truth $\mathbf{a}_{t:t+h}^0$ as a reliable substitute for the corresponding intermediate action chunk during the denoising process.

This gradient guidance $g$ allows us to explicitly calibrate the transitions at any noise level $k$, pushing the VLA's output consistently towards a latent representation that reduces the distance to the target distribution throughout the whole denoising process. As shown in Fig. 2, our method not only steers the pre-trained VLA towards the adaptation data, but also constrains its output to remain within the unified action latent space which happens to be a specific mode of the pre-training action latent space. By preventing the VLAs from deviating far from this structured latent manifold, our method also implicitly helps preserve valuable visuomotor prior knowledge acquired during pre-training. Detailed algorithmic descriptions for how we steer the adaptation of diffusion-based/flow-based VLAs are provided in Algs. 2 and 3.

## 5 EXPERIMENTS

To demonstrate the efficiency of ATE in facilitating the adaptation process across embodiments and tasks, we evaluate our approach in both simulated and real-world settings. Through extensive experiments, our aim is to answer the following key questions: $\mathcal{Q}1$: How efficiently does ATE facilitate the VLA's adaptation under cross-task and cross-embodiment regimes? (Section 5.2) $\mathcal{Q}2$: To what extent does ATE enhance the generalization of the fine-tuned VLA against diverse perturbations, such as variations in object placement, lighting conditions, and visual distractors? (Section 5.3) $\mathcal{Q}3$: How critical is our proposed aligned and structured action latent space to achieving efficient cross-embodiment and cross-task adaptation? (Section 5.4)

### 5.1 EXPERIMENTAL SETUP

**Experimental Environment.** To demonstrate the efficiency of ATE in facilitating the adaptation process across embodiments and tasks, we evaluate our approach in both simulated and real-world settings. For simulation, we evaluate on two benchmarks: **RoboTwin 1.0** (Mu et al., 2025), featuring diverse bimanual tasks, and **ManiSkill3** (Tao et al., 2024), focusing on contact-rich single-arm manipulation. Together, these allow us to test ATE under both bimanual and contact-rich regimes, directly addressing $\mathcal{Q}1$. For real-world experiments, we deploy ATE-trained policies on a dual-arm RealMan robot, with four long-horizon dual-arm tasks and one tool-use task (Appendix B). This setup provides a strong testbed for adaptation across task and embodiment shifts ($\mathcal{Q}1$) and robustness under real-world perturbations such as object placement, lighting, and visual distractors ($\mathcal{Q}2$). Further details are in Appendix D.

**Baselines.** Since ATE is designed to be plug-and-play, we benchmark it on representative diffusion- and flow-matching architectures, including **Diffusion Policy (DP)**, **RDT-1B** (Liu et al., 2024), and $\pi_0$ (Black et al., 2024) (details in Appendix D.1). **Note that in all cases, the baseline comparison is against direct fine-tuning of these architectures without ATE.**

**Implementation.** Firstly, we adopt a two-step Info-VAE training scheme to learn structured latent representations over action chunks. In step-1, Info-VAE is trained on large-scale VLA datasets to learn transferable latent representations, while step-2 trains on small sets of domain-specific data to align the action distribution of the target domain. Detailed dataset configurations and training settings for both steps are provided in Appendix C. Secondly, the pretrained Info-VAE is used to provide structured latent information for a lightweight guidance module, which facilitates efficient adaptation of diffusion- or flow-matching–based VLAs. During policy fine-tuning, the guidance

module leverages the Info-VAE latent space to measure the discrepancy between predicted and target actions and applies gradient signals to steer the policy updates toward the target distribution. The Info-VAE parameters are frozen, with guidance enhanced via random latent perturbations and dynamically scaled according to the current generation step. A tunable parameter controls the overall strength of the guidance, allowing a flexible trade-off between adaptation efficiency and stability. This approach enables rapid adaptation without requiring additional data, and integration into existing VLA training pipelines is straightforward and computationally lightweight. Full algorithmic details, including the latent-space guidance formulation, are provided in the Appendix F.

## 5.2 MAIN RESULTS

**Simulation Results.** To answer $\mathcal{Q}1$, we first evaluate both RDT and $\pi_0$ on 17 manipulation tasks from the RoboTwin 1.0 benchmark, with results summarized in Table 1. Our ATE variants significantly outperform the original models, achieving average success rate improvements of +10% for RDT and +9% for $\pi_0$, with gains exceeding +30% on challenging tasks such as *put apple cabinet* and *empty cup place*. In addition to higher final performance, our method demonstrates improved sample efficiency, surpassing the RDT baseline at just 70k steps compared to 90k. Detailed performance of the DP baseline and ATE variants, showing consistent success rate improvements and accelerated convergence, along with full analysis of in-distribution and out-of-distribution results, is provided in Appendix D.2. The performance gain can be attributed to two key advantages of our algorithm: (i) the unified action latent space, which compactly represents actions of different embodiments and tasks, and bridges the domain gap between pre-training and adaptation; and (ii) the latent space guidance, which explicitly steers fine-tuning towards the target action distribution, which ensures efficient adaptation while preserving valuable visuomotor priors, resulting in both faster convergence and higher final success rates. The effectiveness of the proposed approach is further validated on the ManiSkill3 benchmark using RDT as the policy backbone. Table 5 in Appendix D.2 shows consistent improvements from the ATE adaptation method on two representative tasks, indicating strong generalization in highly dynamic simulated environments. Overall, the results demonstrate that our method consistently enhances performance across tasks and architectures, particularly under limited data conditions.

**Real-World Results.** To further address $\mathcal{Q}1$ in terms of efficient cross-task and cross-embodiment adaptation, we conduct real-world experiments on a dual-arm RealMan robot, with each arm having 7 DoF and equipped with Agibot OminiPicker single-DoF grippers. To be specific, we design four representative manipulation tasks for our experiments. The detailed experimental setups and data collection procedures provided in Appendix D.3 and Appendix I. The experiments focus on long-horizon manipulation sequences, each lasting over one minute and requiring precise coordination between arms and objects, thus providing a strong testbed to validate ATE for efficient cross-task

Table 1: The comparison of task success rates on RoboTwin 1.0 benchmark

| Task | Method | | | |
| --- | --- | --- | --- | --- |
| | RDT-1B (Liu et al., 2024) | **RDT-1B + ATE** (Ours) | $\pi_0$ (Black et al., 2024) | $\pi_0$ + **ATE** (Ours) |
| Block Hammer Beat | 52% | **71%**(↑ 19) | 38% | **44%**(↑ 6) |
| Block Handover | 69% | **91%**(↑ 22) | 80% | **92%**(↑ 12) |
| Blocks Stack (Easy) | 10% | **31%**(↑ 21) | 30% | **50%**(↑ 20) |
| Blocks Stack (Hard) | 1% | **7%**(↑ 6) | **8%** | 7%(↓ 1) |
| Bottle Adjust | **53%** | 37%(↓ 16) | 39% | **45%**(↑ 6) |
| Container Place | 34% | **55%**(↑ 21) | 56% | **59%**(↑ 3) |
| Diverse Bottles Pick | 18% | **24%**(↑ 6) | 20% | **40%**(↑ 20) |
| Dual Bottles Pick (Easy) | 76% | **87%**(↑ 11) | 48% | **85%**(↑ 37) |
| Dual Bottles Pick (Hard) | 39% | **58%**(↑ 19) | 52% | **55%**(↑ 3) |
| Dual Shoes Place | 6% | **9%**(↑ 3) | 22% | **24%**(↑ 2) |
| Empty Cup Place | 22% | **61%**(↑ 39) | 32% | **36 %**(↑ 4) |
| Mug Hanging (Easy) | 6% | 6% | 11% | **27%**(↑ 16) |
| Mug Hanging (Hard) | 1% | 1% | 4% | 4% |
| Pick Apple Messy | 35% | **39%**(↑ 4) | **18%** | 11%(↓ 7) |
| Put Apple Cabinet | 20% | **45%**(↑ 25) | 34% | **55%**(↑ 21) |
| Shoe Place | **44%** | 43%(↓ 1) | 52% | **58%**(↑ 6) |
| Tool Adjust | 54% | 42%(↓ 12) | **70%** | 69%(↓ 1) |
| **Average** | 31.8% | **41.6%**(↑ 9.8) | 36.1% | **44.8%**(↑ 8.7) |

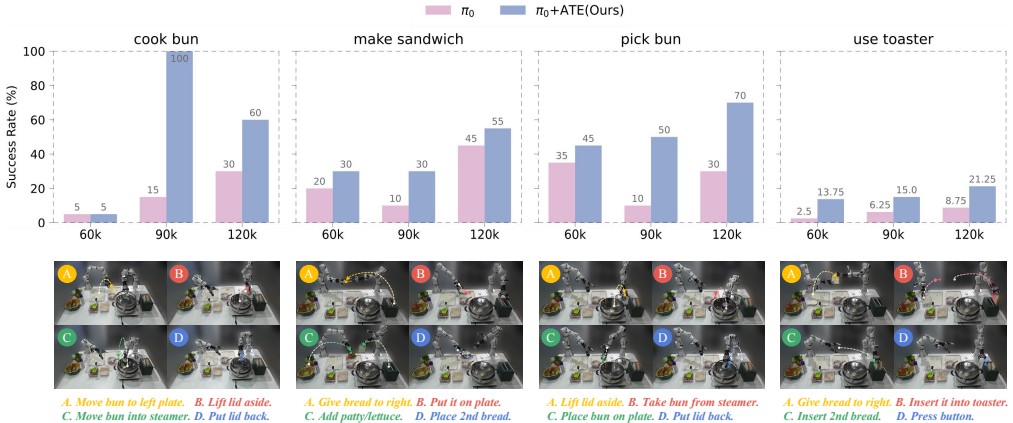

Figure 3: Real-world evaluation on physical robot experiments. Top panel reports success rates across four tasks and overall average, comparing ATE with the baseline $\pi_0$. The success rates are measured at 60k, 90k, and 120k steps to assess both convergence speed and final performance. Bottom panel shows full execution trajectories of four representative tasks, covering long-horizon, single-arm, and dual-arm scenarios.

and cross-embodiment adaptation. The four tasks are *Cook Bun*, *Pick Bun*, *Make Sandwich*, and *Use Toaster*, with detailed definitions provided in Appendix D.3. Figure 3 visualizes success rates across tasks at different training steps. Our method significantly accelerates learning, achieving higher success rates with fewer training steps (detailed analyses are provided in Appendix D.3). Besides, in real-world experiments, we also observe significant improvements in motion quality (see Appendix H for detailed analyses).

## 5.3 GENERALIZATION ABILITY ANALYSIS

To answer $\mathcal{Q}2$, we evaluate the robustness and generalization of ATE under diverse real-world perturbations. Detailed experimental setups, including lighting conditions, spatial shifts, visual distractors, and human disturbances, are provided in Appendix D.4. Each experiment is evaluated over 5 trials, with success defined as completing the full manipulation sequence. The experimental setups for all perturbations are illustrated in Fig. 9, and the corresponding results are reported in Tables 7 and 8, all in Appendix D.4. ATE consistently outperforms the baseline across all generalization experiments, demonstrating stronger robustness and generalization in real-world settings ($\mathcal{Q}2$). This strong generalization arises because our method constrains the policy within the structured latent space learned during pre-training, effectively preserving pre-trained visuomotor priors while allowing adaptation to new environmental variations.

## 5.4 ABLATION STUDY

To answer $\mathcal{Q}3$, we study the impact of the aligned and structured action latent space on cross-embodiment and cross-task adaptation. Our two-step Info-VAE, which embeds adaptation actions into a mode of the pre-trained latent manifold, consistently outperforms a single-step VAE trained directly on task-specific data, using both $\pi_0$ and Diffusion Policy backbones. These results confirm that the aligned latent space is critical for efficient adaptation, enabling downstream policies to rapidly leverage prior knowledge and generalize across different tasks and robotic platforms (see detailed results and the corresponding figure in Appendix D.5).

## 6 CONCLUSION

In this work, we present ATE , a lightweight and data-efficient framework for adapting VLAs to new embodiments and tasks. By unifying disparate action spaces and steering the pre-trained model via classifier guidance, our method effectively bridges the action distribution gap between pre-training and downstream adaptation. The experimental results on representative VLA architectures, including DP, RDT, and $\pi_0$ verify the widely applicability of our adaptation method. The cross-task adaptation experiments on RoboTwin and ManiSkill benchmarks show that ATE consistently outperforms baselines. By constructing complex bimanual collaboration tasks in a new robot platform, we observe ATE obtains significantly better performance than standard adaptation methods, especially for spatial, visual, and disturbance generalization settings. Our results highlight the practicality and scalability of ATE, offering a plug-and-play solution that seamlessly integrates with diffusion-

and flow-based VLAs without altering their architectures. Looking ahead, we envision that this alignment-and-guidance paradigm can serve as a foundation for fine-tuning generalist robot policies at scale, enabling efficient adaptation to new robotic platforms and diverse tasks.

## ETHICS STATEMENT

All procedures in this paper were conducted in accordance with the ICLR Code of Ethics (`https://iclr.cc/public/CodeOfEthics`).

## REPRODUCIBILITY STATEMENT

For the practical part, we give experiment setup in Section 5.1. The hyper-parameters and implementation details are given in Appendix C, D and E. The code is released at `https://github.com/TeleHuman/Align-Then-Steer`.

## ACKNOWLEDGMENTS

This work is supported by the National Natural Science Foundation of China (Grant No.62306242), the Young Elite Scientists Sponsorship Program by CAST (Grant No. 2024QNRC001), and the Yangfan Project of the Shanghai (Grant No.23YF11462200).

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

# A  ADDITIONAL IN-DEPTH DISCUSSIONS

## A.1  ADDITIONAL RELATED WORKS

**Vision-Language-Action Models.** The long-term objective of robotic manipulation is to develop general-purpose models that can adapt to diverse embodiments and tasks. Vision-Language-Action models (VLAs) is currently a mainstream approach that builds upon the perception and language understanding capabilities of pre-trained VLMs and introduces an additional action prediction module. Leveraging massive datasets of robotic demonstrations for joint pretraining, these models can process language instructions, visual observations, and proprioceptive input to generate continuous action sequences that interact with the physical environment to accomplish complex real-world tasks. Early representative works such as Octo and RT-1 (Brohan et al., 2022; Team et al., 2024) trained Transformer-based policies from scratch on large-scale robotic datasets, while RT-2 and OpenVLA (Zitkovich et al., 2023; Kim et al., 2024) further utilized pre-trained VLMs (e.g., PaLI-X, Prismatic) (Chen et al., 2023a; Anschütz & Le Bras, 2023) and fine-tuned them on robot datasets (e.g., OXE) (Collaboration et al., 2023) via action discretization, achieving improved capabilities by inheriting the knowledge of VLMs. Subsequent methods such as RDT-1B, H-RDT, DexVLA, DiVLA, DexGraspVLA, Dita (Liu et al., 2024; Bi et al., 2025; Wen et al., 2025a;c; Zhong et al., 2025; Hou et al., 2025) adopt diffusion-based architectures to represent the multimodality in robot data. Recently, a new generation of VLA models—such as $\pi_0$, GROOT, $\pi_{0.5}$, SmolVLA, and TinyVLA (Black et al., 2024; NVIDIA, 2025; Intelligence et al., 2025; Shukor et al., 2025; Wen et al., 2025b) adopt a dual system compositional architecture that incorporates a generative action expert based on diffusion or flow matching (Liu et al., 2023a; Lipman et al., 2023) to produce continuous action sequences, significantly improving the prediction accuracy and physical executability. However, they mostly focus on architectures and objectives in the pre-training phase, while we aim to improve the efficiency for cross-embodiment and cross-task adaptation on pretrained VLAs.

**Constructing Unified Action Space.** Establishing a shared discrete token dictionary to discretize continuous robotic actions is a common approach for mapping heterogeneous actions into a shared space (Brohan et al., 2022; Zitkovich et al., 2023; Kim et al., 2024; Pertsch et al., 2025). Furthermore, UniVLA (Bu et al., 2025) leveraged a Vector-Quantized VAE to extract task-centric latent actions from massive cross-embodiment videos for VLA pre-training and post-training. Similarly, UniACT (Zheng et al., 2025) also learned a discrete atomic behavior codebook from large-scale robotic data to form a universal discrete latent action representation. However, these methods primarily focus on establishing a universal latent representation for actions across different embodiments, failing to resolve the significant distribution discrepancy that persists between pre-training and adaptation data within the latent space. In contrast, we approach this issue from the perspective of bridging the distribution gap. Our method constructs a unified action space by first encoding the pre-training actions to form a base latent distribution, and subsequently embedding the adaptation actions into the modes of this pre-training action distribution, which results in a structured latent space. While Bauer et al. (2025) combined separate VAEs with contrastive learning to map different embodiment-specific instances of the same action into a shared latent space, their method's heavy reliance on a hand motion retargeting system to generate these instances makes it challenging to apply in common VLA pre-training and post-training recipe.

**More Discussions on Policy Adaptation of VLAs.** Several approaches construct latent action representation for different data sources (Ye et al., 2025; Bu et al., 2025; NVIDIA, 2025), perform Kinematics retargeting or eigengrasps (Bauer et al., 2025; Yuan et al., 2025), or manually design a physically interpretable action space for different robot platforms (Liu et al., 2024; Zheng et al., 2025). While they address the high fine-tuning costs for embodiment and task mismatches, they cannot easily adapt to different robot platforms since the target action distribution can be significantly different from the original one. RoboOS adopts advanced architectures by building a skill library and a shared action space for different embodiments, but it requires complex module construction and coupling. Other works improve the adaptation efficiency of VLAs by caching visual tokens and selectively updating task-relevant ones (Xu et al., 2025), compressing high-frequency action sequences into low-frequency tokens via discrete cosine transform (Pertsch et al., 2025), adopting LoRA-based updates (Wen et al., 2025b), or performing dynamic layer activation for distillation (Zhang et al., 2025a). These methods are often restricted to auto-regressive models and overlook diffusion and flow-matching architectures. Meanwhile, solely increasing training efficiency is still limited in cross-embodiment adaptation as the domain gap would be hard to bridge via direct tuning.

## A.2 RELATIONSHIP TO CFGRL

CFGRL (Frans et al., 2025) establishes a principled link between diffusion-based generative modeling and policy improvement in RL. Specifically, CFGRL formulates the target policy $\pi$ as a product distribution:

$$\pi(a \mid s) \propto \hat{\pi}(a \mid s) \cdot f\big(A_{\hat{\pi}}(s, a)\big),$$

where $\hat{\pi}(a \mid s)$ is a reference (e.g., behavior cloning) policy, $A_{\hat{\pi}}(s, a) = Q_{\hat{\pi}}(s, a) - V_{\hat{\pi}}(s)$ is the advantage function under $\hat{\pi}$, and $f : \mathbb{R} \to \mathbb{R}_{\geq 0}$ is a non-negative, monotonically increasing function. Under this condition, Theorem 1 in the paper proves that $\pi$ guarantees policy improvement: $J(\pi) \geq J(\hat{\pi})$. To enable controllable improvement, CFGRL introduces an exponentiated family of policies parameterized by a guidance weight $w \geq 0$:

$$\pi_w(a \mid s) \propto \hat{\pi}(a \mid s) \cdot f\big(A_{\hat{\pi}}(s, a)\big)^w.$$

This mirrors the role of the inverse temperature $1/\beta$ in KL-regularized RL objectives such as Advantage-Weighted Regression (AWR), where the optimal policy takes the form $\pi(a \mid s) \propto \hat{\pi}(a \mid s) \exp\big(A_{\hat{\pi}}(s, a)/\beta\big)$. However, unlike AWR—where $\beta$ must be fixed during training—CFGRL decouples the reference and optimality components, enabling post-hoc control of $w$ at inference time without retraining.

Sampling from $\pi_w$ is achieved via classifier-free guidance from diffusion or flow models. Let $o \in \{0, 1\}$ denote an optimality label (e.g., $o = 1$ if $A_{\hat{\pi}}(s, a) \geq 0$). The guidance-modulated score (or velocity) field is:

$$\nabla_a \log \pi_w(a \mid s) = (1 - w)\nabla_a \log \hat{\pi}(a \mid s) + w\nabla_a \log \pi(a \mid s, o{=}1),$$

which corresponds to the additive composition of unconditional and optimality-conditioned score functions—a direct consequence of the log-linearity of product distributions. This avoids explicit estimation of the normalizing constant in $p(o \mid s, a)$, and sidesteps instability from backpropagating through learned advantage predictors.

In ATE, the latent-guidance strategy share a common perspective — improving policy behavior by reshaping the action distribution via score-based guidance. However, there exists important conceptual distinctions between ATE and CFGRL. Specifically, CFGRL mainly performs reward-based policy improvement, where the classifier signal is derived from an advantage function $A(s, a)$, and the guided policy provably improves expected returns. In contrast, ATE performs reward-free domain alignment, where the guidance signal is defined solely from target-domain action demonstrations in the unified latent space and does not assume access to rewards or value functions.

But interestingly, CFGRL unlocks hidden gains in goal-conditioned behavioral cloning introduces a goal-conditioned behavioral cloning setting where the classifier term can be interpreted as a proxy for advantage without requiring explicit reward labels. The likelihood $p(g \mid s, a)$ of reaching a goal $g$ from $(s, a)$ inherently satisfies the monotonicity requirement with respect to advantage. Thus, applying CFGRL with $f(A) = p(g \mid s, a)$ and $w > 1$ yields a provably improved policy over standard GCBC, while retaining the simplicity of supervised training. We posit that under idealized assumptions, the distance metric over the unified latent space used in ATE could similarly be viewed as an equivalent proxy for advantage function. With this assumption, ATE can also be rigorously framed as performing one-step policy improvement towards a 'target domain optimality', even without explicit reward labels. Instead of conditioning on a specific goal $g$, we condition on the *Target Domain Manifold* (encapsulating the target embodiment and task). In CFGRL, the optimality variable $o$ implies reaching the goal $g$, while in ATE, the optimality variable $o$ implies aligning with the adaptation data distribution in the unified latent space.

Let the *target domain optimality* $p(o|s, a)$ be proportional to the likelihood that an action $a$ belongs to the target domain optimal action distribution. The assumption suggests this likelihood is expressed by the adaptation-specific VAE encoder $E_\psi$. Assuming RBF kernel parameterization for this likelihood, the probability of an action $a$ matching a target demonstration $a^*$ is:

$$p(o|s, a) \propto \exp\left(-\frac{\|E_\psi(a) - E_\psi(a^*)\|^2}{2\sigma^2}\right),$$

i.e., $f(A(s, a)) \approx \exp\big(-\|E_\psi(a) - E_\psi(a^*)\|^2\big)$, then the guidance term derived in CFGRL (Eq. 9 in [1]) becomes:

$$\nabla_a \log p(o|s, a) \propto -\nabla_a \|E_\psi(a) - E_\psi(a^*)\|^2.$$

In order to apply the guidance from such a optimality indicator, we have to use it during the generation process of score-based model generation process. In other words, $a^k$ would be the noisy action at arbitrary diffusion timesteps $k$, and $a^*$ would thus be the action labels in the demonstration trajectory from the target domain. Since our setup is actually goal-conditioned behavioral cloning setup, we always take the label from the demonstration as the optimal one by default. Then, $\nabla_a \log p(o|s, a)$ *is exactly the guidance term $g$ used in ATE* (Eq. (11)). Under such assumption, ATE is not merely performing domain adaptation; it is mathematically equivalent to performing policy improvement.

## B  ADDITIONAL REAL-WORLD EXPERIMENTS: TOOL USE

### B.1  EXPERIMENTAL SETUP

To validate our method on the tool-use VLA challenge, we collected an additional 80 trajectories and designed the **Make Yogurt Bowl** task. This task requires using both a *shovel* and a *spoon* for precise ingredient manipulation.

**Data.** 80 high-quality trajectories are collected for this task, covering variations in initial object placement.

**Platform.** All experiments are conducted on a dual-arm **RealMan** robot, where each arm has 7 DoF and is equipped with Agibot OminiPicker single-DoF grippers. The left arm handles the spatula, and the right arm handles the spoon.

**Training.** Models are trained for 120k steps on 8 NVIDIA A100 GPUs with a batch size of 12 per GPU. Execution runs at 20 Hz on a single NVIDIA 4090.

**Task Description.** Use the right gripper to pick up the shovel, scoop out the nuts and oats, and then pour yogurt into the bowl. During the process, use the left gripper to stir with a spoon.

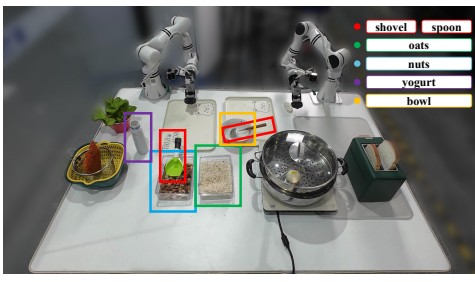

Figure 4: Experimental setup for the *Make Yogurt Bowl* task. The dual-arm robot manipulates ingredients using a spatula (right hand) and a spoon (left hand), demonstrating precise coordination. Key objects are highlighted.

Table 2: Success rate (%).

| Task | Baseline | ATE(Ours) |
|---|---|---|
| Make Yogurt Bowl | 15% | 25% |

### B.2  RESULTS.

For the *Make Yogurt Bowl* task, the robot must perform highly coordinated dual-arm manipulation: the right gripper picks up a spatula to scoop nuts and oats and pour yogurt into the bowl, while the left gripper continuously stirs the mixture with a spoon. This task requires precise control of multiple tools simultaneously, careful timing to avoid spillage, and maintaining proper spatial relationships between ingredients and utensils. Such multi-tool coordination represents a challenging test for VLAs, as small deviations can lead to task failure.

As shown in Table 2, our ATE method achieves a success rate of 25%, outperforming the baseline at 15%, with each method tested over 20 trials. This improvement highlights ATE's ability to leverage pre-trained latent representations and structured visuomotor priors, enabling more reliable

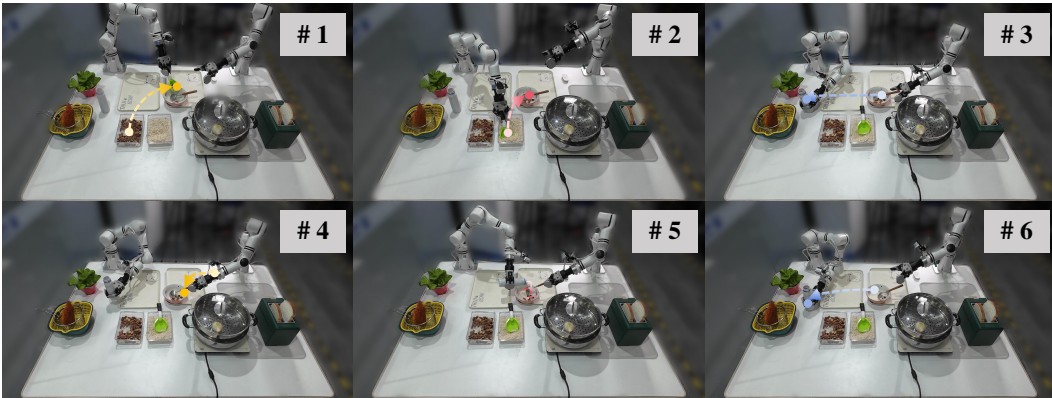

#1: Pick up the shovel with the right gripper. Scoop the nuts using the shovel. #2: Scoop the oats using the shovel. #3: Pick up the yogurt bottle with the right gripper. #4: Pick up the spoon with the left gripper. #5: Pour the yogurt into the bowl while stirring with the spoon in the left gripper. #6: Place the yogurt bottle back in its original position.

Figure 5: Real-world execution of the Make Yogurt Bowl task.

tool handling and fine-grained manipulation. The results indicate that ATE effectively captures the dependencies between dual-arm actions and tool interactions, allowing it to generalize across subtle variations in object positions and motions, whereas the baseline struggles with the complexity of simultaneous tool use.

### B.3 VISUAL DISTRACTOR GENERALIZATION

To evaluate visual distractor robustness, unexpected objects (e.g., extra fruit pieces, small puzzles) are randomly placed on the table. The robot must complete the task while ignoring these unexpected items. Success is defined as correctly completing all phases with proper tool use. Each experiment is repeated 5 times, and the results are summarized in Table 3.

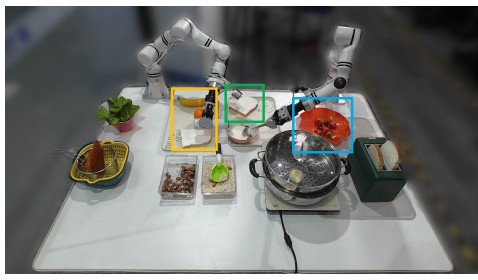

Figure 6: Experimental setup of the *Make Yogurt Bowl* task under visual distractors. Additional irrelevant objects (e.g., fruit pieces) are placed on the table, while the dual-arm robot must focus on manipulating the spoon and spatula to complete the task.

Table 3: Success rate (%) under Visual Distractors.

| Task | Baseline | ATE(Ours) |
|---|---|---|
| Make Yogurt Bowl | 0% | 20% |

## C IMPLEMENTATION DETAILS OF ACTION LATENT ENCODERS

Inspired by prior work on motion sequence VAEs (Petrovich et al., 2021; Chen et al., 2023b), we similarly prepend two learnable tokens, $\mu_{\text{token}}$ and $\Sigma_{\text{token}}$, to the input of our Transformer-based action encoder. The corresponding outputs from the encoder are then used to convert fixed-length

action chunks into a single, chunk-level latent representation. The decoder, built upon a Transformer decoder with a cross-attention mechanism, takes $H$ zero embeddings as a query and the latent vector $z$ as key and value, and generates a reconstructed action chunk of length $H$.

Our two-step training scheme, introduced in Section 5.1, is designed to first learn a general latent structure and then embed actions from specific domains into it. In **Step 1: General Latent Learning**, we train a pre-training VAE on a large-scale VLA dataset to capture a general latent structure across diverse robot platforms and tasks. In **Step 2: Domain-specific Latent Learning**, we train a newly initialized VAE on small sets of domain-specific data to embed the action distribution of the target domain within the latent space learned by the pre-training VAE.

The model uses a latent dimension of 512 and is trained with a mutual information term to ensure informative latent variables. A key advantage of this two-stage approach is its remarkable efficiency. Due to the quantity of the pre-training data, **Step 1** typically requires around 12 hours of training. However, the training in **Step 2** is relatively fast, taking less than 0.5 hours. Table 4 provides the detailed training configurations for our Info-VAE across all policies and stages.

Table 4: Info-VAE training configuration across stages and policy types.

| Policy | Dataset Source | Episodes | Epochs |
|---|---|---|---|
| **Step 1** | | | |
| DP | RoboTwin 1.0: 5 tasks | 50/task | 1000 |
| RDT | Open X-Embodiment (subset), DROID, Kuka, ALOHA | 3000 | 1000 |
| $\pi_0$ | Open X-Embodiment (subset), DROID | 3000 | 1000 |
| **Step 2** | | | |
| DP | RoboTwin 1.0: 1 task | 50/task | 200 |
| RDT | RoboTwin 1.0: 17 tasks, ManiSkill3: 2 tasks | 100/task | 200 |
| $\pi_0$ | RoboTwin 1.0: 17 tasks, Real-World: 4 long-horizon tasks | 50/task | 200 |
| $\pi_0$ | Real-World: 4 long-horizon tasks | 160/task | 200 |

# D    EXPERIMENTAL SUPPLEMENT

## D.1    BASELINE DETAILS

**RDT-1B.** RDT-1B is a diffusion-based VLA model tailored for bimanual manipulation. It leverages a scalable transformer and a physically interpretable unified action space, enabling transfer across different robot embodiments. In our experiments, it is trained and evaluated on multitask RoboTwin 1.0 and ManiSkill3 settings.

Table 5: Task success rates on ManiSkill3.

| Task | RDT-1B | RDT-1B + ATE(Ours) |
|---|---|---|
| Push Cube | 65.2% | **78.4%** (↑ 13.2) |
| Pick Cube | 7.6% | **14.8%** (↑ 7.2) |
| **Average** | 36.4% | **46.6%** (↑ 10.2) |

$\pi_0$. $\pi_0$ is a flow-matching–based generalist policy built on pre-trained vision-language models. It supports multi-task and multi-embodiment manipulation, exhibits strong instruction-following capability, and allows efficient adaptation with limited data. Our implementation uses the Huggingface Lerobot Community version (`https://github.com/huggingface/lerobot`) across all experiments.

**Diffusion Policy (DP).** DP is a foundational diffusion-based architecture commonly used for comparison in VLA and robotic policy benchmarks. It serves as a standard diffusion baseline to evaluate the benefits of `ATE`.

## D.2    FAST ADAPTATION IN ROBOTWIN/MANISKILL

Each model is trained and evaluated across a range of manipulation tasks using the specified number of demonstrations. All experiments are conducted using multi-GPU training setups.

**RoboTwin 1.0 Benchmark**    RoboTwin 1.0 is adopted as the core evaluation benchmark. As a generative digital twin framework tailored for dual-arm tasks, RoboTwin 1.0 integrates 3D generative models and large language models to synthesize diverse object-centric scenes along with spatially grounded robot code. A total of 17 tasks are selected, spanning both single-arm and dual-arm manipulation scenarios (e.g., tool adjustment, dual-bottle picking). Besides, RoboTwin 1.0 provides 17 diverse single- and dual-arm manipulation tasks that are largely absent from mainstream VLA pretraining datasets (Open X-Embodiment, Droid, etc.). Together, this benchmark allows us to evaluate ATE under bimanual regimes, directly addressing $\mathcal{Q}1$ regarding cross-task efficient adaptation.

**ManiSkill3 Benchmark**    In addition, we use ManiSkill3, a fast, GPU-parallelized simulator designed for contact-rich single-arm manipulation. This simulator emphasizes contact-rich manipulation and offers a diverse suite of single-arm tasks. We evaluate our method on 2 representative tasks such as push cube and pick cube, which emphasize physical contact and fine-grained control.

**Simulation Training Configuration.** We summarize the training settings for each policy in Table 6.

Table 6: Training configurations for different policy models and environments. BS/GPU denotes batch size per GPU.

| Policy | Benchmark | BS/GPU | GPUs | Training Duration | Dataset Size |
|---|---|---|---|---|---|
| RDT | RoboTwin 1.0 (17 tasks, multi-task) | 16 | 4 A100 | 100k steps | 100/task |
| | ManiSkill3 (2 tasks, multi-task) | 12 | 2 A100 | 100k steps | 100/task |
| $\pi_0$ (PyTorch) | RoboTwin 1.0 (17 tasks, multi-task) | 6 | 4 A100 | 60k steps | 50/task |
| DP | RoboTwin 1.0 (5-task pretrain) | 128 | 1 A100 | 2k epochs | 50/task |
| | RoboTwin 1.0 (Fine-tune) | 128 | 1 A100 | 300 epochs | 50/task |

**Diffusion Policy Evaluation.** To assess the effectiveness of our proposed ATE training strategy, we perform evaluations under both in-distribution and out-of-distribution settings using Diffusion Policy as the backbone model:

- **Pretraining:** A base model is trained on a fixed set of 5 RoboTwin 1.0 tasks, including block handover, empty cup place, put apple cabinet, blocks stack easy, and blocks stack hard.

- **Evaluation:**

    - *In-distribution setting:* We fine-tune and evaluate the model on a subset of the same tasks used in pretraining to assess in-distribution task adaptation.

    - *Out-of-distribution setting:* We fine-tune and evaluate the model on novel RoboTwin 1.0 tasks that were not part of the pretraining set, measuring generalization to unseen tasks.

- **Discussion:** Fig. 7 illustrates the performance of both the DP baseline and the ATE variant in in-distribution and out-of-distribution settings. Overall, our method consistently improves success rates and accelerates convergence, with particularly large gains on tasks where the baseline struggles. Examining the training curves at 100, 200, and 300 epochs shows that our method reaches higher success rates earlier and maintains strong performance on the most challenging tasks.

This setup allows us to verify whether our ATE policy fine-tuning improves generalization and adaptation performance across both familiar and novel tasks.

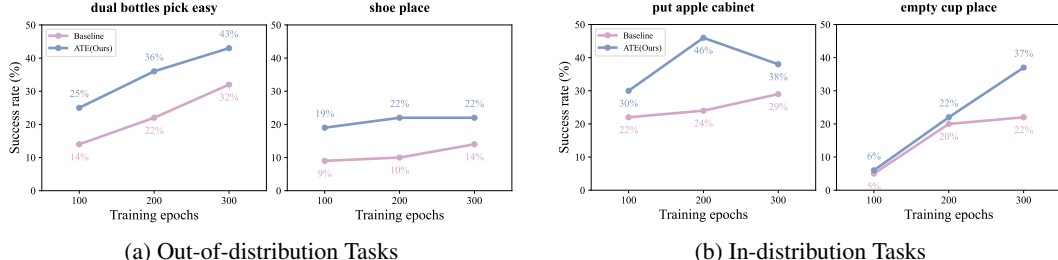

Figure 7: Evaluation of DP baseline with and without ATE across (a) Out-of-distribution tasks and (b) In-distribution tasks. Incorporating ATE consistently improves success rates and accelerates convergence, with the most significant gains observed on challenging tasks where baseline performance is low.

### D.3 FAST ADAPTATION TO DUAL REALMAN ARMS IN REAL WORLD

**Experimental Goals.** These evaluations aim to measure adaptation efficiency across task and embodiment shifts ($\mathcal{Q}1$) and to assess robustness to real-world perturbations such as object placement, lighting, and visual distractors ($\mathcal{Q}2$).

**Platform and Robot.** All experiments are conducted on a dual-arm **RealMan** robot, where each arm has 7 DoF and is equipped with Agibot OminiPicker single-DoF grippers. *Note that mainstream VLA pretraining datasets (e.g., RDT, $\pi_0$) predominantly use 6-DoF single-arm robots, highlighting the increased embodiment complexity in RealMan.*

**Vision Setup.** The visual system consists of a main overhead camera (Intel RealSense L515, RGB only, 960×540, 30 Hz, automatic exposure) and two wrist-mounted cameras (Intel RealSense D405, RGB only, 960×540, 30 Hz, automatic exposure) to capture the workspace from multiple perspectives.

**Data Collection.** We collect 160 high-quality trajectories for each of the four representative long-horizon manipulation tasks (*Cook Bun, Pick Bun, Make Sandwich, Use Toaster*), each lasting over one minute and requiring precise bimanual coordination between arms and objects.

**Discussion.** Figure 3 in Section 5.2 visualizes success rates across tasks at different training steps. Our method significantly accelerates learning, achieving higher success rates with fewer training steps. For example, in *Cook Bun*, the success rate increases to 100% in 90k steps, while the baseline only reaches 15%. In the single-arm *Pick Bun* task, ATE reaches 50% success at 90k steps and 70% at 120k. For dual-arm tasks (*Make Sandwich* and *Use Toaster*), ATE consistently improves performance across all checkpoints. On average across all tasks, the baseline achieves 16.7% overall success, while ATE reaches 58.1% at 120k steps, demonstrating substantial improvement in both final performance and convergence speed. The largest gains are observed in long-horizon tasks like *Pick Bun*, where proper sequencing and coordination are crucial. Besides, in real-world evaluation, we also observe manifest improvements in motion quality. Here, we use the *Cook Bun* task as a representative example, with all related 6-dimensional force data and analyses provided in Appendix H.

**Training.** Models are trained for 120k steps using 8 NVIDIA A100 GPUs, with a batch size of 6 per GPU.

**Task Setup and Success Criteria.** These tasks require precise bimanual coordination, tool interaction, and long-horizon reasoning, posing significant challenges for existing VLA models. We design four long-horizon manipulation tasks and one tool-use task:

- **Cook Bun:** The right gripper grasps a bun from the right plate and places it on the left plate, while the left gripper lifts the lid and places it aside. The left gripper then picks a bun from the left plate and places it into the steamer, and finally returns the lid to the steamer.
- **Pick Bun:** The left gripper lifts the lid temporarily, grasps the bun farthest from the camera inside the steamer, and places it on the left plate. The lid is then returned to the steamer.

- **Make Sandwich:** The left gripper retrieves one toasted bread slice from the toaster and transfers it to the right gripper, which places it on the left plate. The right gripper then adds the patty, lettuce, and top bread slice handed from the left gripper.

- **Use Toaster (Tool-Use Task):** The left gripper picks up the farther bread slice and passes it to the right gripper for toaster insertion. The remaining slice is inserted similarly, followed by pressing the toaster button. This task specifically evaluates tool-use capabilities.

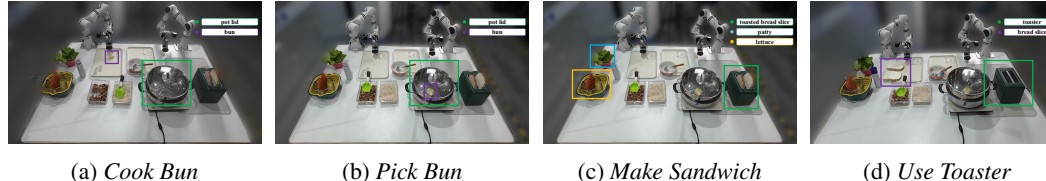

(a) *Cook Bun*  (b) *Pick Bun*  (c) *Make Sandwich*  (d) *Use Toaster*

Figure 8: Key objects involved in the four representative long-horizon manipulation tasks.

**Evaluation.** The evaluation is run on a single NVIDIA 4090 at 20 Hz, and each task is evaluated over 20 trials. Success is defined as completing the long-horizon sequence, including coordinated object manipulation and placement. Initial object positions are randomly sampled within a radius of 3.5 cm. For the *Use Toaster* task, we adopt a graded success metric: fully inserting a bread slice counts as 1.0, tilted insertion as 0.25, and partial insertion as 0.5.

### D.4 GENERALIZATION ABILITY EVALUATION ON DUAL REALMAN ARMS

This section provides the real-world generalization experiment setups as follows, covering illumination variations, spatial shifts, visual distractors, and human disturbances, as illustrated in Figures 9 and 10, with the corresponding success rates summarized in Tables 7 and 8. Overall, across all generalization experiments—including varying illumination, visual distractors, spatial shifts, human disturbances, and object morphology/clutter variations—ATE consistently outperforms the baseline. These results directly address $\mathcal{Q}2$, showing that ATE enhances the robustness of the VLA against diverse real-world perturbations. The strong generalization arises because our method constrains the policy within the structured latent space learned during pre-training, which effectively preserves pre-trained visuomotor priors while allowing adaptation to new environmental variations. By maintaining behavior within this latent manifold, ATE enables the policy to recover from unexpected disturbances, focus on task-relevant features, and adapt to unseen object configurations, demonstrating reliable performance under conditions not encountered during training.

Table 7: Success rates (%) under different illumination settings (15 trials per task).

| Illumination | Method | Cook Bun | Pick Bun | Make Sandwich |
|---|---|---|---|---|
| Low | $\pi_0$ | 0% | 0% | 40% |
| | $\boldsymbol{\pi_0} + \texttt{ATE}$ | **60%** | **40%** | 40% |
| High | $\pi_0$ | 0% | 0% | 40% |
| | $\boldsymbol{\pi_0} + \texttt{ATE}$ | **60%** | **40%** | **60%** |
| Flickering | $\pi_0$ | 0% | 0% | 20% |
| | $\boldsymbol{\pi_0} + \texttt{ATE}$ | **20%** | 0% | 20% |

Table 8: Success rates (%) under different generalization challenges.

| Challenge | Cook Bun | | Pick Bun | | Make Sandwich | |
|---|---|---|---|---|---|---|
| | $\boldsymbol{\pi_0} + \textbf{ATE}$ | $\pi_0$ | $\boldsymbol{\pi_0} + \textbf{ATE}$ | $\pi_0$ | $\boldsymbol{\pi_0} + \textbf{ATE}$ | $\pi_0$ |
| Visual Distractor | **45%** | 30% | **75%** | 30% | **50%** | 40% |
| Spatial Generalization | **40%** | 0% | **60%** | 40% | **40%** | 20% |
| Human Disturbance | **5%** | 0% | **45%** | 10% | **50%** | 40% |

**Varying Illumination.** We alter the lighting conditions while keeping the color temperature fixed at 5800 K. The illumination levels are set to low, high, and flicker around the optimal value. As shown in Table 7, ATE consistently outperform the baseline, particularly under extreme or flickering illumination, where the baseline often failed. This indicates that ATE is robust to lighting intensity

variations, maintaining task performance even under conditions not seen during training. We used a set of controllable studio lamps to vary lighting conditions. All lamps have a fixed color temperature of 5800 K. All illumination settings are illustrated in Figure 10. Each illumination condition (low, high, flickering) was evaluated with 5 trials per task, resulting in 15 trials per task in total.

- **Lamp:** COB studio fill light
- **Low Illumination:** 25 lux
- **High Illumination:** 65 lux
- **Flickering Illumination:** alternating between 0 and 45 lux, flicker frequency 2 Hz

**Spatial Generalization.** To further examine position generalization, we extend the randomized initial placement experiments beyond the original 20 trials. The objects are displaced within a larger radius of approximately 6.5 cm, with five trials conducted at each boundary position (up, down, left, right) as well as at the initial location. For the *Pick Bun* and *Cook Bun* tasks, the buns were moved accordingly, while for the *Make Sandwich* task, the meat slices and lettuce were repositioned. Table 8 shows that ATE achieved success rates of $40\%$, $60\%$, and $40\%$ on *Cook Bun*, *Pick Bun*, and *Make Sandwich*, respectively. These results demonstrate ATE's ability to adapt to larger spatial shifts, highlighting efficient position generalization across diverse manipulation tasks. As illustrated in Figure 9, each of the five locations was evaluated with five trials. We first compute a success rate per location, and the reported result is the mean success rate across the five locations.

**Visual Distractor.** We introduce irrelevant objects that are not present in the dataset, such as scattered fruits, puzzle pieces, and trays, to place on the table as visual distractions. Table 8 shows that ATE outperformed the baseline in all tasks, with success rates of $40\%$ for *Cook Bun*, $80\%$ for *Pick Bun*, and $40\%$ for *Make Sandwich*. This suggests that ATE effectively focuses on task-relevant objects while ignoring irrelevant clutter, demonstrating strong visual generalization. Each task was evaluated over 20 trials.

**Human Disturbance.** During the task, when the robotic arm successfully grasped a bun or other objects, a human quickly interfered—for example, removing a bun from the gripper or repositioning ingredients—forcing the policy to reattempt. Table 8 shows that ATE retained $0\%$ success on *Cook Bun*, $40\%$ on *Pick Bun*, and $60\%$ on *Make Sandwich*, whereas the baseline largely failed. These results highlight ATE's robustness to sudden external interventions, suggesting that the policy can recover from unexpected disturbances and adapt its behavior accordingly. Each task was evaluated over 20 trials.

### D.5    ABLATION STUDY

To further analyze the impact of the aligned and structured action latent space, we conduct a detailed ablation study comparing three adaptation strategies:

1. **Direct fine-tuning**: The original policy ($\pi_0$ or Diffusion Policy) is fine-tuned directly on task-specific adaptation data without using ATE.
2. **ATE (Single-step Info-VAE)**: Adaptation is performed using ATE with a single-step Info-VAE trained directly on task-specific data, removing the two-step procedure.
3. **ATE (Two-step Info-VAE)**: Full adaptation using our proposed ATE to fine-tune the policy.

**Results**    Figure 11 presents the comparison across these three strategies. The results show that:

- Direct fine-tuning achieves lowest success rates.
- ATE with single-step Info-VAE performs similarly to direct fine-tuning, with minor improvements, indicating that merely using ATE without the structured latent space provides limited benefits.
- ATE with two-step Info-VAE consistently outperforms both alternatives, achieving higher final success rates. The two-stage latent space effectively reduces the domain gap while preserving pre-trained visuomotor priors, enabling more efficient and robust cross-task and cross-embodiment adaptation.

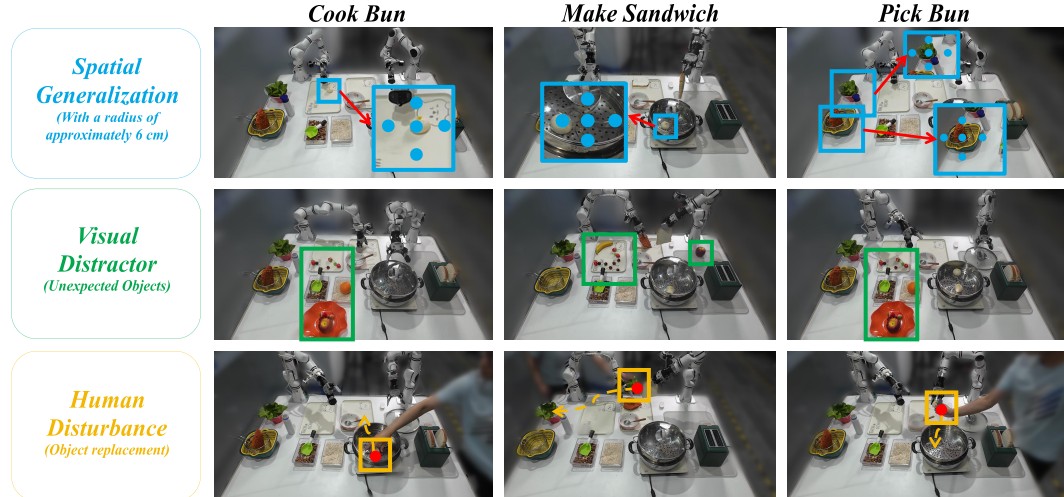

Figure 9: Real-world generalization settings. For the Spatial Generalization experiment, we vary the initial positions of manipulated objects. For the Visual Distractors experiment, we randomly place unseen items (e.g., fruits) in the scene. For the human disturbance experiment, we intervene at key execution points by resetting already completed sub-tasks.

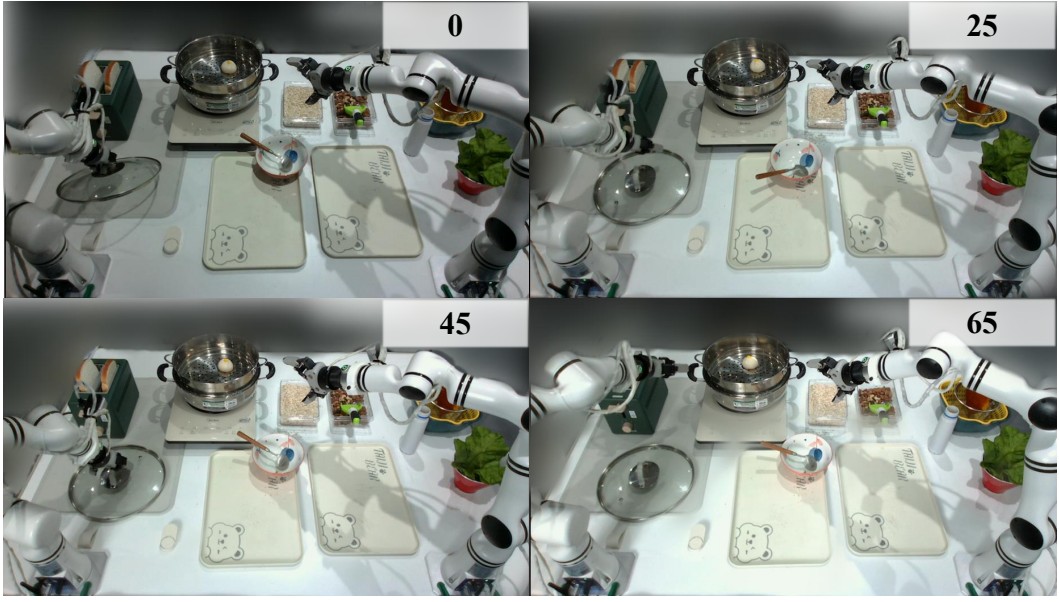

Figure 10: Illustration of the illumination setup under three conditions: low (25 lux), high (65 lux), and flickering (alternating between 0 and 45 lux at 2 Hz).

## D.6 OVERALL DISCUSSION

Our simulation and real-world experiments consistently demonstrate that ATE facilitates efficient cross-task and cross-embodiment adaptation. Across both the RoboTwin 1.0 and ManiSkill3 benchmarks, our framework improves final success rates, accelerates convergence, and enhances sample efficiency compared to baseline policies. Real-world evaluations on the dual-arm RealMan robot

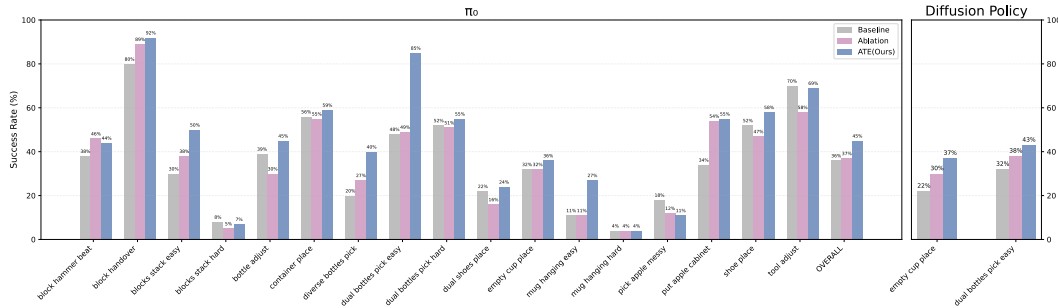

Figure 11: Ablation study on the effect of two-step Info-VAE training. We compare the proposed two-step Info-VAE with a single-step VAE trained only on task-specific datasets, using both $\pi_0$ and Diffusion Policy backbones. For the Diffusion Policy results, we evaluate both in-distribution (†) and out-of-distribution generalization settings (‡).

further confirm these advantages, with `ATE` achieving higher success rates, smoother trajectories, better long-horizon coordination, and stronger generalization.

These results indicate that our approach effectively addresses $\mathcal{Q}1$ by leveraging a unified action latent space and latent-space guidance. This combination allows the policy to rapidly adapt to new tasks and embodiments while preserving valuable pre-trained visuomotor knowledge, significantly enhancing the practicality and deployment of pre-trained VLA models.

# E  TRAINING HYPERPARAMETERS

We summarize the key hyperparameter settings used for both stages of our method, including (i) Learning the Unified Action Latent Space, and (ii) Steering Efficient Adaptation with Latent Guidance.

Table 9: Hyperparameters for Info-VAE pretraining.

| Hyperparameter | Value |
|---|---|
| Latent dimension | 512 |
| Temporal input length | 64 (RDT), 50 ($\pi_0$), 14 (DP) |
| Input dimensionality | 128 (RDT), 32 ($\pi_0$), 14 (DP) |
| Optimizer | Adam |
| Learning rate | $1 \times 10^{-4}$ |
| Weight decay | $1 \times 10^{-4}$ |
| Batch size | 64 |

Table 10: Task-specific fine-tuning (RoboTwin).

| Hyperparameter | RDT | $\pi_0$ | DP |
|---|---|---|---|
| Chunk size | 64 | 50 | 8 |
| Batch size | 64 | 24 | 128 |
| Learning rate | 1.0e-4 | 2.5e-5 | 1.0e-4 |
| Training duration | 100k steps | 60k steps | 300 epochs |

**Other platforms.**  For ManiSkill and Real Robot experiments, we use the following task-specific fine-tuning settings:

- **ManiSkill (simulator):** Chunk size 64 (RDT), 24 batch size, learning rate 1.0e-4, training duration 100k steps.

- **Real Robot (physical):** Chunk size 20 ($\pi_0$), batch size 48, learning rate 2.5e-5, training duration 120k steps.

# F  ALGORITHMIC DESCRIPTION OF ATE

---

**Algorithm 1** Stage 1: Learning the Unified Action Latent Space

---

**Input:** Pre-training dataset-$\mathcal{D}_{\text{pretrain}}$, Adaptation dataset $\mathcal{D}_{\text{adaptation}}$, Pre-training action VAE $\phi$, Adaptation action VAE $\psi$, Number of epochs $N$
   // Learn the pre-training action VAE
**for** epoch $= 1$ to $N$ **do**
      Sample a batch of actions $\{\mathbf{a}_{t:t+H-1}^{(i)}\}_{i=1}^{n}$ from $\mathcal{D}_{\text{pretrain}}$
      Update pre-training action VAE $\phi$ via maximizing $\mathcal{L}(\phi; \mathcal{D}_{\text{pretrain}})$ in Eq. (4)
**end for**
   // Calculate the pre-training action latent distribution
Initialize the running mean $\mu_\phi$ and variance $\Sigma_\phi$ of the pre-trained action latent distribution $q_\phi(z)$
**for** all possible sample $\mathbf{a}_{t:t+H-1}$ in **do**
      Encode the latent $z$ of the action chunk $\mathbf{a}_{t:t+H-1}$ via $\phi$
      Update the running mean $\mu_\phi$ and variance $\Sigma_\phi$
**end for**
   // Learn the adaptation action VAE
**for** epoch $= 1$ to $N$ **do**
      Sample a batch of actions $\{\mathbf{a}_{t:t+L-1}^{(i)}\}_{i=1}^{n}$ from $\mathcal{D}_{\text{adaptation}}$
      Update pre-training action VAE $\psi$ via maximizing $\mathcal{L}(\psi; \mathcal{D}_{\text{adaptation}}, \phi)$ in Eq. (5)
**end for**

---

**Algorithm 2** Stage 2: Steering Adaptation of **Flow-based VLAs** with Latent Guidance

---

**Input:** Flow-based VLA $v_\theta$, Adaptation dataset $\mathcal{D}_{\text{adaptation}}$, Adaptation action encoder $\mathbf{E}_\psi$, Guidance scale $\lambda$, Learning rate $\eta$
**repeat**
      Sample an observation-language-action pair $(\mathbf{o}_t, l, \mathbf{a}_{t:t+L-1}) \sim p_{\mathcal{D}_{\text{adaptation}}}(\cdot)$
      Sample $\epsilon \sim \mathcal{N}(0, I)$, $\tau \sim \mathcal{U}([0, 1])$
      Corrupt clean action chunk $\mathbf{a}_{t:t+L-1}^{\tau} = \tau \mathbf{a}_{t:t+L-1} + (1 - \tau)\epsilon$
      // Compute the guidance
      Get the latents $\hat{z} = \mathbf{E}_\psi(\mathbf{a}_{t:t+L-1}^{\tau})$, $z = \mathbf{E}_\psi(\mathbf{a}_{t:t+L-1})$ of noisy action chunk and clean action chunk
      Obtain the latent guidance $g = -\nabla_{\mathbf{a}_{t:t+L-1}^{\tau}} \|z - \hat{z}\|^2$
      // Steer the fine-tuning
      Get the ground truth velocity $v_t = \mathbf{a}_{t:t+L-1} - \epsilon$
      Update the VLA via $\theta = \theta - \eta \nabla_\theta \|[v_\theta(\mathbf{a}_{t:t+L-1}^{\tau}; \tau, \mathbf{o}_t, l) + \frac{1-\tau}{\tau}\lambda g] - v_t\|^2$
**until** Converged

---

**Algorithm 3** Stage 2: Steering Adaptation of **Diffusion-based VLAs** with Latent Guidance

---

**Input:** Diffusion-based VLA $\epsilon_\theta$, Adaptation dataset $\mathcal{D}_{\text{adaptation}}$, Adaptation action encoder $\mathbf{E}_\psi$, Guidance scale $\lambda$, Learning rate $\eta$
**repeat**
      Sample an observation-language-action pair $(\mathbf{o}_t, l, \mathbf{a}_{t:t+L-1}) \sim p_{\mathcal{D}_{\text{adaptation}}}(\cdot)$
      Sample $\epsilon \sim \mathcal{N}(0, I)$, $k \sim \mathcal{U}(\{1, 2, \ldots, T\})$
      Corrupt clean action chunk $\mathbf{a}_{t:t+L-1}^{k} = \sqrt{\bar{\alpha}_k}\mathbf{a}_{t:t+L-1} + \sqrt{1 - \bar{\alpha}_k}\epsilon$
      // Compute the guidance
      Get the latents $\hat{z} = \mathbf{E}_\psi(\mathbf{a}_{t:t+L-1}^{k})$, $z = \mathbf{E}_\psi(\mathbf{a}_{t:t+L-1})$ of noisy action chunk and clean action chunk
      Obtain the latent guidance $g = -\nabla_{\mathbf{a}_{t:t+L-1}^{k}} \|z - \hat{z}\|^2$
      // Steer the fine-tuning
      Get the ground truth velocity $v_t = \mathbf{a}_{t:t+L-1} - \epsilon$
      Update the VLA via $\theta = \theta - \eta \nabla_\theta \|[\epsilon_\theta(\mathbf{a}_{t:t+L-1}^{k}; k, \mathbf{o}_t, l) - \sqrt{1 - \bar{\alpha}_k}\lambda g] - \epsilon\|^2$
**until** Converged

---

| Parameter | Value (%) |
|---|---|
| DoF | 8 × 2 = 16 |
| Arm weight | 7.9 × 2 = 15.8 kg |
| Arm Payload | 5000g |
| Arm reach | 638.5mm |
| Arm Repeatability | 0.05mm |
| Joint motion range | J1 ± 178°, J2 ± 130°, J3 ± 178°, J4 ± 135°, J5 ± 178°, J6 ± 128°, J7 ± 360° |
| Gripper range | 120mm |
| Gripper weight | 0.43kg |
| Gripper Payload | 1.5kg |
| Gripper max force | 30N |

Table 11: Technical specifications. In this table, the term arm refers to the Realman RM75-6s model, while gripper refers to the Ominipicker developed by AGIBOT.

## G  HARDWARE DETAILS

We present the dual-arm platform constructed and utilized in this work. Two identical Realman RM75-6s robotic arms are mounted on a white lifting desk using four G-type fixtures. At their end-effectors, we fixed the OminiPicker grippers developed by AGIBOT together with RealSense-D405 short-range depth cameras. Both grippers communicate with the workstation through USB-to-serial protocols. In addition to the two cameras mounted on the robotic arm end-effectors, we also strategically positioned one RealSense-L515 high-precision depth camera on a tripod, positioned appropriately behind the arms to capture a complementary global view. All three cameras are directly connected to our workstation via USB 3.2 interfaces. The two robotic arms are connected to the same local area network as the workstation and communicate using the Ethernet-based SDK provided by Realman. An overview of the hardware setup is illustrated in Figure 12, while the corresponding hardware specifications are listed in Table 11 for reference.

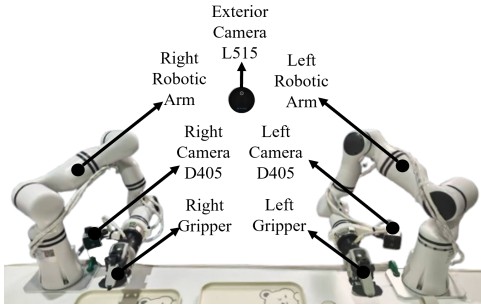

Figure 12: Hardware feature. The wrist-mounted camera is the RealSense D405 model, with an optimal operating range of 7cm – 50cm, while the exterior camera is the RealSense L515 model, with an optimal operating range of 0.25m – 9m. The two robotic arms are of exactly the same model rather than mirror versions.

# H    THE HIDDEN ADVANTAGE OF ATE: SAFER AND SMOOTHER MANIPULATION

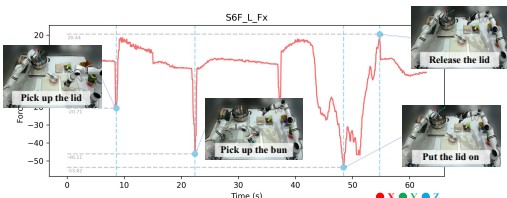

(a) 6-axis force data during a single Cook Bun task using the direct fine-tuned $\pi_0$ model.

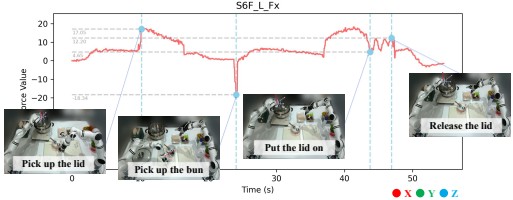

(b) 6-axis force data during a single Cook Bun task using the $\pi_0$ model fine-tuned with ATE .

Figure 13: **Six-axis force comparison during the *Cook Bun* task.** (a) Using the direct fine-tuned $\pi_0$ model, the force signals exhibit instability and less structured patterns. (b) Using the $\pi_0$ model fine-tuned with ATE, the force signals are smoother and more consistent, indicating improved robustness and better adaptation to the target embodiment.

While our primary objective was to facilitate data-efficient adaptation by bridging the action distribution gap, our experiments revealed **a valuable and unanticipated benefit**: superior motion smoothness and more stable force control. We validate this finding through a detailed analysis of six-axis force measurements. The term six-axis force refers to forces and torques expressed in the coordinate frame rigidly attached to the target object: the three force components are along the positive directions of the coordinate axes, while the remaining three correspond to torques around these axes.

When performing the cook bun task with the direct fine-tuned policy $\pi_0$, we observed that the robot arm frequently exerts excessive force while placing the steam lid back on the steamer, which often led to deformation of the steamer. By contrast, when the same action was executed by the fine-tuned policy $\pi_0$ with ATE, the deformation of the steamer was significantly mitigated.

To quantitatively substantiate this observation, we used the built-in six-axis force sensor integrated into the robot's end-effector to collect data during task execution. As shown in Figure 13a and Figure 13b, we selected the force component along the x-axis as the primary evaluation criterion. This is because we confirmed that, when placing the lid, the force transmitted from the lid to the gripper is aligned with the x-axis direction. The data presented in the figure are relative measures, obtained by subtracting the sensor readings at the initial pose from the raw sensor values.

From the figures, it can be observed that the force curve of ATE  exhibits fewer sharp fluctuations compared to that of the baseline and remains at a smaller overall magnitude. More specifically, across four representative peaks and valleys, the baseline policy demonstrates a tendency to apply excessive and unnecessary downward force during the bun-grasping and lid-placing actions.

This finding suggests that by constraining the fine-tuning process within the pre-training action latent space, our method implicitly guides the policy toward safer and more robust physical interactions.

# I    REAL-WORLD DATA COLLECTION DETAILS

In this section, we provide a detailed description of how our dataset is collected. As illustrated in Figure 14b, our real-world tasks involve bimanual simultaneous actions. If we were to rely on traditional single-operator teleoperation, accomplishing such tasks would impose excessive demands on the operator. To address this challenge, we adopt a dual-operator collaborative teleoperation approach for data collection.

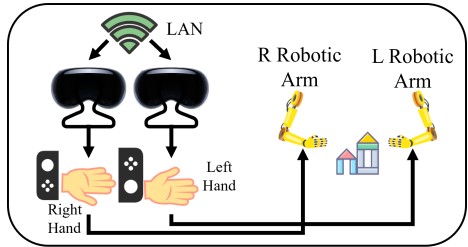
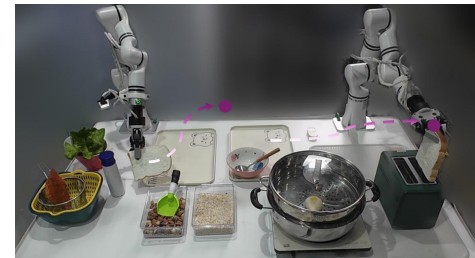

(a) Dual-arm collaborative teleoperation.          (b) Bimanual task involves simultaneous actions.

Figure 14: The setup of real-robot data collection. (a) The overview of how we implement dual-arm data Collection. During teleoperation, the operator holds Joycon with the hand which not cared about by our program. (b) An example that requires both hands to perform logically disparate tasks, simultaneously occurs in the *use toaster* task.

In the dual-operator teleoperation setup, each operator is first equipped with a separate Vision Pro headset, both of which are connected to the same local area network. We then employ the open-source project TrackingStreamer to extract hand-keypoint estimations from the Vision Pro devices and stream them to our workstation. Note that in the wrist pose estimated by Vision Pro, the attached coordinate frame on the wrist differs in orientation from the coordinate frame defined on the Real-man robotic arm. To ensure consistency, we apply a right-multiplication with a predefined rotation matrix $T_{\text{rotation}}$ to the wrist frame, so that when both the wrist and the robotic arm are in their initial poses, the directions of their coordinate frames are aligned.

After both operators place their corresponding hands (which control the robotic arms) in the initial positions and press the initialization button, the program immediately records the wrist estimation from Vision Pro at that moment, and treats it as the initialization matrix for the current data collection session. Next, we assume the existence of a transformation matrix, such that left-multiplying the initialization matrix with $T_{\text{transform}}$ maps it to the robotic arm coordinate system, perfectly aligning it with the initial pose of the end-effector . By right-multiplying both sides of this equation with the inverse of the initialization matrix, we can solve for $T_{\text{transform}}$

$$T_{\text{transform}} = T_{\text{ee\_init}}(T_{\text{vision\_init}}T_{\text{rotation}})^{-1}$$

where $T_{\text{ee\_init}}$ denotes the initial pose of the end effector in the robot coordinate system, and $T_{\text{vision\_init}}$ denotes the wrist pose estimated by Vision Pro at the time of initialization. For every subsequent step, Vision Pro's wrist estimation is first multiplied by the right $T_{\text{rotation}}$ and then by the left $T_{\text{transform}}$. This way, the motion and orientation of the hand can be directly mapped into the robotic arm coordinate system, representing the desired end effector pose. By employing this mapping approach for teleoperation, it is only necessary to know the initial poses of the robotic arm and the initial pose of the interface to be used as the teleoperation reference. Moreover, under the requirements of our experiments, this method enables a clear separation of the two different robotic arms, thereby facilitating the realization of dual operator teleoperation. Next, we utilized the Pinocchio library to solve the inverse kinematics to get the joint angles, which are then transmitted directly to the robot for execution. Beyond using Vision Pro as our primary sensing hardware, we integrate Nintendo Joy-Con devices: operators can use the Joy-Cons to control the gripper's open and close actions, thereby completely eliminating potential misrecognition issues since we use vision-based solutions. Additionally, Joy-Cons serve as auxiliary controllers to notify the operator of code errors, quickly set the robot arms, and confirm dataset recording, among other functions.

The advantages of employing dual operator teleoperation can be outlined as follows. First, tasks requiring both arms to perform different actions simultaneously can be executed effortlessly, enabling the policy which fine-tuned on our dataset to demonstrate superhuman multitasking capabilities. Second, when one arm is idle, its wrist-mounted camera can be aimed at the other working arm, thereby providing three distinct visual perspectives for the active robotic arm. Third, operators are granted greater freedom of movement: they can walk around to achieve larger control ranges for

their respective arms without worrying about the state of the other arm. Furthermore, since the two operators' hands are spatially separated, the visual occlusion problem in HandOver-type tasks is naturally resolved. Finally, each of the operator only needs to master half of the skill set, which accelerates skill transfer when switching to new tasks and reduces operator workload.

# J ADDITIONAL EXPERIMENTS

## J.1 ADDITIONAL EXPERIMENTS: SAMPLE EFFICIENCY UNDER VARYING FINE-TUNING DATASET SCALES

To explicitly evaluate the sample efficiency of our method, we compared the success rates of the $\pi_0$ backbone with and without ATE on the RoboTwin 1.0 benchmark across varying fine-tuning dataset sizes: 10, 25, and 50 episodes per task. The results are summarized in Table 12.

Table 12: Comparison of sample efficiency on RoboTwin 1.0 tasks. ATE demonstrates significant improvements in low-data regimes. Notably, with **25 episodes**, ATE achieves a **3.15×** improvement in average success rate (0.290 vs. 0.092).

| Task | 10 episodes / task | | 25 episodes / task | | 50 episodes / task | |
|---|---|---|---|---|---|---|
| | $\pi_0$ | $\pi_0$ + **ATE** | $\pi_0$ | $\pi_0$ + **ATE** | $\pi_0$ | $\pi_0$ + **ATE** |
| Put Apple Cabinet | 0% | **0%**(↑ 0) | 2% | **27%**(↑ 25) | 34% | **55%**(↑ 21) |
| Blocks Stack (Easy) | 0% | **1%**(↑ 1) | 5% | **6%**(↑ 1) | 30% | **50%**(↑ 20) |
| Diverse Bottles Pick | 7% | **7%**(↑ 0) | 4% | **22%**(↑ 18) | 20% | **40%**(↑ 20) |
| Dual Bottles Pick (Easy) | 5% | **14%**(↑ 9) | 21% | **59%**(↑ 38) | 48% | **85%**(↑ 37) |
| Block Handover | 1% | **14%**(↑ 13) | 31% | **31%**(↑ 0) | 80% | **92%**(↑12) |
| **Average** | 2.6% | **4.6%**(↑ 2) | 9.2% | **29.0%**(↑ 19.8) | 42.4% | **64.4%**(↑ 22) |

As shown in Table 12, direct fine-tuning of $\pi_0$ struggles significantly in data-scarce regimes. Notably, with only 25 episodes, the baseline fails to acquire meaningful behaviors for complex tasks (e.g., 2% on *Put Apple Cabinet*), whereas ATE achieves a 27% success rate. Furthermore, ATE with 25 episodes often matches or exceeds the performance of the baseline trained with 50 episodes (e.g., 0.59 vs. 0.48 on *Dual Bottles Pick Easy*), suggesting an approximate 2× improvement in sample efficiency.

## J.2 ADDITIONAL EXPERIMENTS: EFFECTS OF VAE GUIDANCE SCALE

The guidance scale $\lambda$ in Eq. (7) and Eq. (10) controls the strength of the steering signal during the fine-tuning process. We conducted a sensitivity analysis on the $\pi_0$ backbone to identify the optimal operating range.

Table 13: Ablation study on the guidance scale parameter $\lambda$. Evaluation performed on selected RoboTwin tasks.

| **Task** | $\pi_0$ (Baseline) | $\lambda = 1$ | $\lambda = 2$ (**Ours**) | $\lambda = 3$ |
|---|---|---|---|---|
| Mug Hanging (Easy) | 11% | 21%(↑ 10) | **27%** (↑ 16) | 16%(↑ 5) |
| Tool Adjust | **70%** | 58%(↓ 12) | 69%(↓ 1) | 65%(↓ 5) |
| Shoe Place | 52% | 56%(↑ 4) | **58%** (↑ 6) | 56%(↑ 4) |
| Dual Bottles Pick (Easy) | 48% | 82%(↑ 34) | **85%**(↑ 37) | 84%(↑ 36) |

Table 13 indicates that $\lambda = 2$ generally yields the best performance. While stronger guidance ($\lambda = 3$) can enforce the target distribution more aggressively, it may occasionally disrupt the smoothness of the action chunk generation, leading to slightly degraded performance compared to the optimal setting.

## J.3 Additional Experiments: Effects of Reverse KL Weight

The parameter $\alpha$ governs the weight of the reverse KL divergence term in the adaptation VAE objective in Eq. (5), which determines the strength of the mode-seeking behavior during latent alignment.

Table 14: Ablation study on the reverse KL penalty weight $\alpha$. Evaluation performed on selected RoboTwin tasks.

| Task | $\pi_0$ (Baseline) | $\alpha = 0.1$ | $\alpha = 1$ (Ours) | $\alpha = 10$ |
|---|---|---|---|---|
| Blocks Stack Easy (Easy) | 30% | 35%(↑ 5) | **50%**(↑ 20) | 26%(↓ 4) |
| Diverse Bottles Pick | 20% | 20%(↑ 0) | **40%**(↑ 20) | 10%(↓ 10) |
| Block Handover | 80% | **97%**(↑ 17) | 92%(↑ 12) | 37%(↓ 43) |
| Dual Bottles Pick (Easy) | 48% | 60%(↑ 12) | **85%**(↑ 37) | 43%(↓ 5) |

As observed in Table 14, an appropriately tuned $\alpha$ ($\alpha = 1$) is crucial. When $\alpha$ is too large ($\alpha = 10$), the adaptation VAE becomes over-constrained to the pre-training prior, losing the expressivity required to represent the target embodiment's specific actions. Conversely, a small $\alpha$ ($\alpha = 0.1$) results in weak alignment, making the guidance signal less effective.

## J.4 Additional Experiments: Ablation Study on VAE choices

Our steering module is not constrained to InfoVAE; it is designed as a plug-and-play guidance mechanism that can operate on any latent space with meaningful structure.

To verify this generality, we replaced InfoVAE with a **vanilla VAE** while keeping our alignment-learning stage and steering module unchanged. The results are summarized as Table 15 and Table 16.

Table 15: Comparison of different latent structures for guiding Diffusion Policy. InfoVAE consistently outperforms Vanilla VAE due to better structured latent spaces.

| Task | Method | | |
|---|---|---|---|
| | DP (Baseline) | + Vanilla VAE | + InfoVAE (Ours) |
| Dual Bottles Pick (Easy) | 32% | 39% (↑ 7) | **43%**(↑ 11) |
| Shoe Place | 14% | 18% (↑ 4) | **22%**(↑ 8) |
| Put Apple Cabinet | 29% | 42% (↑ 13) | **46%**(↑ 17) |
| Empty Cup Place | 22% | 34% (↑ 12) | **37%**(↑ 15) |

Table 16: Comparison of different latent structures for guiding $\pi_0$. InfoVAE consistently outperforms Vanilla VAE due to better structured latent spaces.

| Task | Method | | |
|---|---|---|---|
| | $\pi_0$ (Baseline) | + Vanilla VAE | + InfoVAE (Ours) |
| Blocks Stack (Easy) | 30% | 31%(↑ 1) | **50%**(↑ 20) |
| Diverse Bottles Pick | 20% | 36%(↑ 16) | **40%**(↑ 20) |
| Dual Bottles Pick (Easy) | 48% | 81%(↑ 33) | **85%**(↑ 37) |
| Block Handover | 80% | 94%(↑ 14) | **92%**(↑ 12) |

These findings confirm three important properties of our design:

- **General compatibility**: The steering module consistently improves the success rate even when paired with a standard VAE, demonstrating that it directly applies to other latent-based methods without architectural changes.
- **Why InfoVAE performs better**: Standard VAEs commonly suffer from inaccurate amortized latent inference, leading to poorly structured latent representations. This effect prevents reliable cross-domain latent alignment, which is essential for correct steering in the target domain. InfoVAE alleviates this problem by enforcing stronger mutual-information constraints, forming a more structural latent structure. Thus, steering becomes more stable and effective.

- **Modularity as a design choice**: The key contribution of our approach lies in factorizing latent space alignment and downstream steering. Better latent models naturally bring larger gains, making our approach adoptable with stronger latent learning methods.

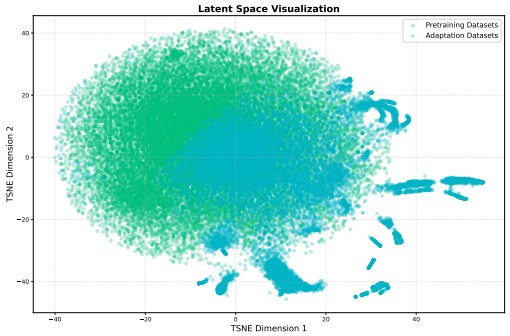 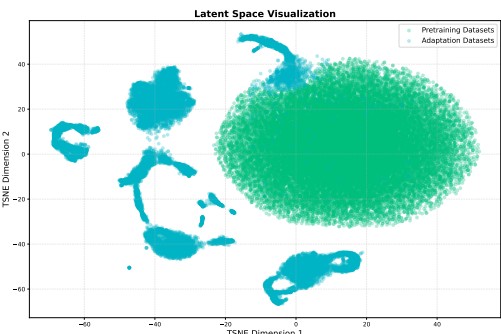

(a) t-SNE of latent representations with InfoVAE.     (b) t-SNE of latent representations with Vanilla VAE.

Figure 15: **Latent Space Visualization.** We project the latent embeddings of both pre-training (green) and adaptation (blue) datasets using t-SNE. Compared to Vanilla VAE (right), InfoVAE (left) produces a significantly more compact and structured latent space, where the adaptation data distribution is tightly aligned with the high-density modes of the pre-training distribution. This alignment is critical for effective cross-domain knowledge transfer.

To further qualitatively validate these claims, we visualized the latent distributions of both pre-training and adaptation datasets using t-SNE, as shown in Figure 15. The visualization clearly demonstrates that **InfoVAE** produces a compact, well-structured latent manifold where the adaptation data (blue points) are embedded closely within the modes of the pre-training data (green points). In contrast, the **Vanilla VAE** results in a loosely aligned latent space with noticeable gaps between the two distributions. This visual evidence supports our quantitative results, confirming that the superior performance of InfoVAE stems from its ability to enforce better distributional alignment, which is a prerequisite for effective steering during fine-tuning.

## J.5 ADDITIONAL EXPERIMENTS: COMPARISON AGAINST OTHER BASELINES

We compared ATE with two related guidance-based methods: DSRL (Wagenmaker et al., 2025) (an RL-based approach) and DynaGuide (Du & Song, 2025) (a test-time steering approach).

**Comparison with Offline DSRL.** We evaluated the methods on the LIBERO-10 task suite. To adapt DSRL for this setting, we implemented it as an offline RL method using the adaptation dataset as a static buffer with sparse rewards for successful trajectories.

Table 17: Comparison with Offline DSRL on the LIBERO-10 benchmark. DSRL degrades performance in this setting, whereas ATE improves it.

| Benchmark | Directly Fine-tuned $\pi_0$ | + Offline DSRL | + ATE (Ours) |
|---|---|---|---|
| LIBERO 10 | 0.78 | 0.43 | **0.88** |

As shown in Table 17, DSRL significantly degrades performance ($0.78 \rightarrow 0.43$). This confirms that RL-based objectives can be unstable when learning from limited expert data without online interaction, suffering from the out-of-distribution issue during value estimation. In contrast, ATE's augmented objective preserves the nature of supervised learning and thus robustly improves performance ($0.88$).

**Comparison with DynaGuide.** We compared Diffusion Policy (DP) baselines against DynaGuide and ATE on three tasks. For a fair comparison, we modified DynaGuide to support multi-view inputs.

Table 18: Comparison with DynaGuide on specific tasks. ATE provides consistent improvements and is more stable.

| Task (DP Backbone) | Base Policy | + DynaGuide | + ATE (Ours) |
|---|---|---|---|
| Put Apple Cabinet | 0.29 | 0.46 | **0.46** |
| Empty Cup Place | 0.22 | 0.30 | **0.37** |
| Dual Bottles Pick Easy | 0.32 | 0.18 | **0.43** |

Table 18 shows that while DynaGuide improves performance on some tasks, it exhibits instability on others (e.g., degrading performance on *Dual Bottles Pick*). Furthermore, we observed that DynaGuide severely impacts inference speed due to iterative gradient calculation during generation, making it less suitable for real-time control compared to ATE, which incurs zero inference overhead.

## K    LLM USAGE STATEMENT

In accordance with ICLR 2026 policy, we report that a large language model (LLM) was used to aid and polish the writing of this manuscript. The authors reviewed all LLM-generated suggestions and take full responsibility for the final content of the paper.

