# OpenReview forum: "Align-Then-stEer: Adapting the Vision-Language Action Models through Unified Latent Guidance"
_ICLR.cc/2026/Conference — ICLR 2026 Poster_

### Official Review · Reviewer_RFTf · 2025-10-31

**Soundness:** 3
**Presentation:** 3
**Contribution:** 3
**Rating:** 6
**Confidence:** 4

**Summary:**

This paper presents Align-Then-Steer (ATE), a framework for adapting pre-trained Vision-Language-Action (VLA) models to new embodiments and tasks.
ATE addresses the action-space mismatch problem when transferring VLAs across robots or manipulation domains. It consists of two stages:
(1) Align – construct a unified latent action space by aligning source and target action distributions using a dual-VAE with reverse KL regularization;
(2) Steer – apply latent guidance to direct diffusion or flow-based VLA generation toward the aligned latent manifold during fine-tuning.
This approach enhances adaptation efficiency, preserves pre-trained capabilities, and achieves strong results in both simulation and real-world settings.

**Strengths:**

1. The motivation is clear and well grounded.

2. The method shows significant improvements in adaptation success rates.

3. The innovation is moderate, but the approach is scalable and extensible.

4. The paper is well organized and easy to follow, with clear figures and mathematical formulations.

**Weaknesses:**

See questions.

**Questions:**

1. I recently came across latent action learning in a survey paper [1]. How does the latent action learning proposed in your work differ from that?

2. This question is more discussion-oriented — I also believe latent learning is a promising approach for cross-embodiment transfer. What other potential solutions do you foresee in the future? And compared to [2], what advantages does your method offer?

3. If there is a large gap between pre-training and fine-tuning data, will this alignment still be effective? How robust is the latent alignment when the target robot has a significantly different kinematic structure or action semantics?

4. How does the reverse KL constraint affect convergence? Did you observe mode collapse or over-constrained behavior during training?

5. What is the model scale, and can it run in real time for control tasks?

6. Are there visualizations (e.g., t-SNE plots) showing how the latent distributions align before and after training?

7. Possibly due to the page limit, some ablation studies are placed in the appendix. However, this is not ideal — key experimental results should be presented in the main text.

[1] Towards a Unified Understanding of Robot Manipulation: A Comprehensive Survey. arXiv 2025.

[2] villa-X: Enhancing Latent Action Modeling in Vision-Language-Action Models. arXiv 2025.

---

> ### Author Response · Authors · 2025-11-25
> **Response to Reviewer RFTf (Part 1)**
>
> Dear Reviewer RFTf,
>
> We sincerely appreciate your precious time and constructive comments. In the following, we would like to answer your concerns separately.
>
> **Question 1**: I recently came across latent action learning in a survey paper [1]. How does the latent action learning proposed in your work differ from that?
>
> **Response**: Thank you for your insightful question and for pointing us to the recent survey [1]. As summarized in [1], existing methods in latent action-space learning broadly fall into two categories: those employing discrete representations and those using continuous ones. Our method aligns with the latter but introduces a fundamentally different objective and architectural design. While most prior works—such as those cited in [1]—use latent actions primarily to (i) provide compact, dynamics-aware action primitives or (ii) serve as inputs to a latent policy, our framework does not learn a policy directly in the latent action space. Instead, we leverage the latent space to quantify the discrepancy between the generated actions and the target action distribution, enabling the construction of a guidance signal that facilitates cross-domain generalization of a pre-trained visuomotor policy.
>
> A particularly relevant comparison is with LAD [2], which also addresses cross-embodiment generalization by aligning diverse embodiments (e.g., human hands, anthropomorphic robots, and parallel grippers) into a shared latent action space. However, LAD assumes access to data from all source and target embodiments during training, allowing it to learn a unified encoder–decoder that aligns across domains a priori. In contrast, our setting is more challenging: the target domain’s embodiment and action space are entirely unknown at training time. Consequently, a single shared latent space (as in LAD) is not feasible.
>
> To address this, ATE trains separate adaptation-specific VAEs for the pre-trained (source) and target (adaptation) domains. Crucially, we impose a KL-divergence constraint that encourages the adaptation latent distribution to be embedded within a high-probability mode of the pre-trained latent prior. This design enables a structured initialization of the policy in the unknown target domain, effectively steering the pre-trained VLA policy toward feasible behaviors without requiring target-domain action labels or embodiment knowledge during training.
>
> **Question 2**: This question is more discussion-oriented — I also believe latent learning is a promising approach for cross-embodiment transfer. What other potential solutions do you foresee in the future? And compared to [3], what advantages does your method offer?
>
> **Response**: Thank you for this thoughtful and forward-looking question. We agree that latent learning holds great promise for cross-embodiment transfer, and we’re glad to discuss both the broader landscape and the specific advantages of our approach relative to [3].
>
> Prior work such as [3] (building on foundations like LAPA [4]) leverages forward and inverse dynamics models to learn a latent action space that is dynamics-consistent—i.e., capable of predicting future states and reconstructing actions from state transitions. This paradigm has been successfully adopted in large-scale VLA systems like GR00T [5] and GO-1 [6], particularly to bridge the dynamics gap between human demonstration videos and robot execution. By aligning human and robot trajectories in a shared latent action space, these methods enable unified pretraining on large-scale in-the-wild human data.
>
> However, this dynamics-based design comes with assumptions and overhead that are mismatched with our goal. Our setting assumes both pretraining and adaptation data are collected from robotic systems (i.e., no human videos). Consequently, the primary challenge is not reconciling fundamentally different dynamics (e.g., human vs. robot), but adapting a fixed pre-trained VLA policy to a novel, unknown robot embodiment with minimal computation and no retraining of the base policy. This leads to two key limitations of adopting [3]-style dynamics modeling in our context: 1) Computational incompatibility: Different VLAs (e.g., DP, π₀, RDT) output actions in heterogeneous formats and dimensions. Learning a universal dynamics model over high-dimensional image observation or proprioceptive states for each would require significant per-policy customization and expensive forward-model training. 2) Distributional rigidity: Dynamics-constrained latent actions tend to concentrate around deterministic transitions, resulting in overly centralized latent distributions. This makes it difficult to embed the target-domain latent distribution into a compatible mode of the pre-trained prior—precisely the mechanism our method (ATE) relies on for adaptation.

---

> > ### Author Response · Authors · 2025-11-25
> > **Response to Reviewer RFTf (Part 2)**
> >
> > In contrast, ATE learns a lightweight, dynamics-agnostic latent action space via action-only VAEs. This design is intentionally minimal: it captures action distribution shifts between domains without modeling state transitions. Crucially, the latent space naturally supports discrepancy quantification (via energy-based function), which we use to construct a guidance signal that steers the pre-trained policy in the target domain. This enables plug-and-play adaptation across diverse VLAs without modifying their architecture or requiring dynamics modeling.
> >
> > Looking ahead, we believe building hybrid approaches that combine coarse dynamics-free alignment (like ours) with on-the-fly dynamics refinement during deployment. But for the current problem of efficient adaptation to unseen embodiments, ATE’s simplicity, compatibility, and distribution-aware guidance offer distinct practical advantages over dynamics-intensive alternatives like [3].
> >
> > -----
> >
> > **Question 3**: If there is a large gap between pre-training and fine-tuning data, will this alignment still be effective? How robust is the latent alignment when the target robot has a significantly different kinematic structure or action semantics?
> >
> > **Response**: Thank you for raising this important and insightful question regarding the robustness of our alignment strategy under significant distributional shifts. Our alignment mechanism is designed under the realistic assumption that modern pre-trained VLA are trained on large-scale, cross-embodiment datasets (e.g., Open X-Embodiment, DROID) that already span a wide variety of kinematic structures (e.g., 6-DoF arms, 7-DoF anthropomorphic hands, parallel grippers) and task semantics. Consequently, the pre-trained action distribution often covers a broad region of the underlying action manifold, even if not densely populated in every sub-region. During adaptation, the target embodiment typically exhibits a more concentrated action distribution (due to limited task scope and fixed morphology). Our two-stage VAE design—particularly the reverse KL constraint in the adaptation VAE—exploits this structure by embedding the target action distribution into a mode of the pre-trained latent prior. This mode-seeking behavior ensures that even if the pre-training data did not include the exact target robot, the latent alignment remains feasible as long as the target actions lie within the convex hull or semantic neighborhood of the pre-trained action space.
> >
> > That said, in extreme cases—for instance, adapting a policy trained solely on single-arm planar manipulators to a full humanoid with 20+ DoFs and fundamentally different action semantics, the pre-trained latent space may indeed lack sufficient representational capacity. In such scenarios, the alignment may become less precise, requiring the guidance signal in the diffusion/flow-matching stage to exert stronger correction over more denoising steps to pull the generated actions toward the target distribution. Nevertheless, the pre-trained VLA still provides a strong inductive bias—semantic understanding, scene grounding, and coarse visuomotor priors—that is far more informative than training from scratch. As evidenced in our real-world experiments (Section 5.2), even when shifting from standard single-arm pre-training (e.g., Droid dataset) to a dual-arm 14-DoF RealMan robot, ATE achieves much higher success rates than direct fine-tuning, demonstrating practical robustness.
> >
> > Additionally, we conducted a "toy setup" experiment. We pre-trained a Diffusion Policy on the Dual-Arm RoboTwin dataset and then fine-tuned it for a Single-Arm task (*Push Cube* in ManiSkill). This represents a significant structural mismatch (collapsing 14-DoF semantics to 8-DoF). The result is reported as follows.
> >
> > | Method                | Pre-training Domain | Target Domain          | Success Rate (50 trials) | Improvement |
> > |-----------------------|---------------------|------------------------|--------------------------|-------------|
> > | DP w/o ATE            | Dual-Arm (RoboTwin) | Single-Arm (ManiSkill) | 36 / 50 (72%)            | -           |
> > | DP w/ ATE (Ours)       | Dual-Arm (RoboTwin) | Single-Arm (ManiSkill) | 41 / 50 (82%)            | +10%        |
> >
> > Despite the significant kinematic gap, ATE achieves an 82% success rate, outperforming direct fine-tuning by 10%. This demonstrates that ATE can effectively identify and align with the relevant subspace (the single-arm mode) within the complex dual-arm prior.
> >
> > To conclude, while extreme cases present challenges, our results indicate that ATE gracefully handles significant kinematic shifts. By leveraging the rich semantic priors of the frozen VLA and the mode-seeking property of our alignment, ATE remains more data-efficient and generalizable than naive fine-tuning.

---

> > > ### Author Response · Authors · 2025-11-25
> > > **Response to Reviewer RFTf (Part 3)**
> > >
> > > **Question 4**: How does the reverse KL constraint affect convergence? Did you observe mode collapse or over-constrained behavior during training?
> > >
> > > **Response**: Thank you for this insightful question regarding the role and practical impact of the reverse KL constraint in our framework. As noted in Section 4.1 and supported by classical variational inference theory [7], the reverse KL divergence $D_{\mathrm{KL}}(q_\psi(z \mid \tilde{a}) \,\|\, q_\phi(z))$ induces mode-seeking behavior. This ensures that the adaptation action latent distribution $q_\psi(z)$—learned from the limited target-domain data—is embedded into a high-probability mode of the pre-trained latent distribution $q_\phi(z)$, thereby constructing a unified and compact latent space $\mathcal{Z}$. We add additional experiments to study this problem. The results are listed as follows.
> > >
> > > | Task                   | $\pi_0$    | $\pi_0$ + ATE w/ $\alpha=1$ (Ours) | $\pi_0$ + ATE w/ $\alpha=10$ | $\pi_0$ + ATE w/ $\alpha=0.1$ |
> > > |-----------------------|-------|--------------------|-------------|--------------|
> > > | blocks stack easy      | 0.30  | **0.50**           | 0.26        | 0.35         |
> > > | diverse bottles pick   | 0.20  | **0.40**           | 0.10        | 0.20         |
> > > | block handover         | 0.80  | 0.92               | 0.37        | **0.97**     |
> > > | dual bottles pick easy | 0.48  | **0.85**             | 0.43        | 0.60         |
> > >
> > > | Task | $\pi_0$ |$\pi_0$ + ATE w/ $\lambda=2$ (Ours) | $\pi_0$ + ATE w/ $\lambda=1$ | $\pi_0$ + ATE w/ $\lambda=3$ |
> > > |--------------------|------|-----------------------------|----------------------|----------------------|
> > > | mug hanging easy   | 0.11 | **0.27**                        | 0.21                 | 0.16                 |
> > > | tool adjust        | **0.70** | 0.69                        | 0.58                 | 0.65                 |
> > > | shoe place         | 0.52 | **0.58**                        | 0.56                 | 0.56                 |
> > > | dual bottles pick easy | 0.48 | **0.85**                        | 0.82                | 0.84                 |
> > >
> > > According to the above table, we observed that the scale of the reverse KL penalty indeed affects the convergence and downstream adaptation performance.
> > > - When the KL weight is too large, the adaptation VAE becomes over-constrained: it prioritizes staying within the pre-trained latent prior at the expense of accurately reconstructing or representing the target-domain action distribution. Although the latent codes remain well-aligned with the pre-trained space, the encoder loses expressivity for the target embodiment, leading to degraded policy performance.
> > > - Conversely, when the KL weight is too small, the alignment between $q_\psi$ and $q_\phi$ becomes weak. The two latent spaces fail to overlap sufficiently, which undermines the guidance signal in the steering stage (Section 4.2). As a result, the classifier guidance—computed as the gradient of the latent distance $\|E_\psi(\hat{a}) - E_\psi(a^*)\|^2$—becomes misaligned with the pre-trained VLA’s native action manifold, yielding unreliable or noisy steering and slower or unstable convergence. Through systematic ablation (see Appendix D.5 and hyperparameter sweeps in Appendix E), we identified a sweet spot where the reverse KL provides sufficient alignment without sacrificing the adaptation VAE’s ability to capture target-domain structure.
> > >
> > > In all reported experiments, we use this balanced setting, which consistently enabled stable training, effective guidance, and improved sample efficiency—evidenced by both simulation and real-world results (Sections 5.2–5.3). Importantly, we did not observe severe mode collapse (e.g., all latents collapsing to a single point) in our experiments. This is attributed to our use of InfoVAE, which includes a mutual information maximization term to prevent posterior collapse and ensure informative latent representations. The reverse KL is a practically tunable mechanism that governs the trade-off between alignment and expressivity, and its proper calibration is essential to ATE.
> > >
> > > **Question 5**: What is the model scale, and can it run in real time for control tasks?
> > >
> > > **Response**: Thanks for the question. We would like to address your concern from the following three perspectives.
> > >
> > > - **Model Scale**: Our ATE framework introduces an asymmetric dual-VAE with a total of approximately **52M parameters**. Importantly, the VAE modules are **used only during training** for latent alignment and providing guidance signal, i.e., **they are discarded at inference time**. Thus, the deployed policy has the exact same computational footprint as the original pre-trained VLA model (e.g., $\pi_0$ or RDT), without any additional latency or memory requirements.

---

> > > > ### Author Response · Authors · 2025-11-25
> > > > **Response to Reviewer RFTf (Part 4)**
> > > >
> > > > - **Real-Time Robot Control**: Yes, our method supports real-time closed-loop control. We have successfully deployed $\pi_0$ fine-tuned with ATE, on a **dual RealMan Robotic Arms platform** in real-world experiments, demonstrating stable manipulation behavior (details in Appendix D.3).
> > > >
> > > > - **Control Frequency**: Through parallelized inference and motor control, we achieve a stable control frequency of **20 Hz**. While the system can reach **~50 Hz**, we found that **maintaining 20 Hz** — matching the data collection rate — yields **more reliable action prediction** and reduces control vibration in real-world execution.
> > > >
> > > > In summary, ATE maintains the same computational cost as the base VLA policy and can be deployed for real-time control without any additional hardware or latency overhead.
> > > >
> > > > -----
> > > >
> > > > **Question 6**: Are there visualizations (e.g., t-SNE plots) showing how the latent distributions align before and after training?
> > > >
> > > > **Response**:  Thank you for the excellent suggestion. We have added visualization results in the revised manuscript (Fig. 15 in Appendix J.4).
> > > > Specifically, we compare latent embeddings of both pre-training and adaptation datasets using **Vanilla VAE** and **InfoVAE** via t-SNE projections.
> > > >
> > > > As shown in the figures:
> > > > - **Vanilla VAE** produces a loosely structured latent space with noticeable distributional mismatch between the pre-training action domain and the adaptation action domain.
> > > > - **InfoVAE** yields a significantly more compact and semantically clustered latent structure, where adaptation data is closely aligned with the high-density modes of pre-training data.
> > > >
> > > > These visualizations confirm that **stronger latent regularization enables more effective cross-embodiment alignment**, validating the necessity of our latent modeling choices in ATE.
> > > >
> > > >
> > > > **Question 7**: Possibly due to the page limit, some ablation studies are placed in the appendix. However, this is not ideal — key experimental results should be presented in the main text.
> > > >
> > > > **Response**: Thank you for this valuable feedback. We fully agree that ablation studies play a central role in validating the design choices of our method. Due to the strict page limit imposed by the conference (8 pages for main content), we prioritized space for: a clear exposition of the two-stage ATE framework (Section 4); comprehensive main results across simulation and real-world cross-embodiment settings (Section 5.2); and the robustness and generalization analyses under realistic perturbations (Section 5.3). Consequently, the detailed ablations such as two-step VAE design vs. single-step alternatives (which directly validates the necessity of our alignment strategy) was moved to Appendix D.5. However, we acknowledge that this may reduce its visibility. In the next version, we will revise the main text to include a concise version of this ablation—for instance, as a compact table or a short paragraph in Section 5.4—while preserving all current content through minor rephrasing and tightened presentation. This will ensure that the core evidence supporting our key technical contribution remains readily accessible to readers.
> > > >
> > > > -----
> > > >
> > > > **Reference**
> > > >
> > > > [1] Towards a Unified Understanding of Robot Manipulation: A Comprehensive Survey. arXiv 2025.
> > > >
> > > > [2] Bauer E, Nava E, Katzschmann R K. Latent Action Diffusion for Cross-Embodiment Manipulation. arXiv:2506.14608, 2025.
> > > >
> > > > [3] villa-X: Enhancing Latent Action Modeling in Vision-Language-Action Models. arXiv 2025.
> > > >
> > > > [4] Ye S, Jang J, Jeon B, et al. Latent Action Pretraining from Videos. ICLR 2025
> > > >
> > > > [5] Bjorck J, Castañeda F, Cherniadev N, et al. Gr00t n1: An open foundation model for generalist humanoid robots. arXiv:2503.14734, 2025.
> > > >
> > > > [6] Bu Q, Cai J, Chen L, et al. Agibot world colosseo: A large-scale manipulation platform for scalable and intelligent embodied systems. arXiv:2503.06669, 2025.
> > > >
> > > > [7] Bishop, Christopher M., and Nasser M. Nasrabadi. Pattern recognition and machine learning. Vol. 4, no. 4. New York: springer, 2006.

---

### Official Review · Reviewer_XTcT · 2025-10-31

**Soundness:** 3
**Presentation:** 2
**Contribution:** 3
**Rating:** 6
**Confidence:** 3

**Summary:**

Align then Steer (ATE) provides a sample-efficient manner to adapt into a new embodiment for pretrained Vision-Language-Actions Models. ATE first trains a VAE that learns a latent space of all actions in the pretraining dataset, $z$, and uses it as a reference for the adaptation dataset. Afterwards, the method then trains another VAE to learn a separate latent for the adaptation dataset. The novelty came from aligning the two distributions together using a reverse KL penalty. Afterwards, for effective policy adaptation, one can use classifier guidance as an additional objective in the adaptation loss.

Via both real-world and simulated benchmarks, ATE achieved considerable improvements in performance across dexterous tasks across various forms of disturbances and difficulties as well as faster convergence.

**Strengths:**

1. The authors conducted an extensive suite of real-world and simulated experiments, which does bolster the claim that the paper is making. I like that the author made a lot of validation on the robustness of the experiments by checking ATE against various forms of environment perturbations.
2. I believe that this type of alignment procedure in the finetuning stage is also doing something similar to that of RL. I believe that you can follow the same line of thinking proposed in CFGRL [1] and show that this form of classifier guidance in finetuning is a form of policy improvement.

References:

[1] Frans, K. et al., 2025. “Diffusion Guidance Is a Controllable Policy Improvement Operator”. ArXiv.

**Weaknesses:**

1. While the paper shows considerable improvement in aggregate, each task is only evaluated with 5 trials (correct me if I’m wrong). If possible, it is better to use a larger number of trials per task (10-20) so that there is less variance in your experiment procedure.
2. The method section is pretty dense in notation. I believe that the paper can be reformatted for better readability.

**Questions:**

1. How does ATE compare to other guidance methods such as DSRL or DynaGuide (although their formulations are different)?
2. Can authors also discuss how the sample efficiency of this method stack up compared to the baseline methods across various finetuning set sizes? I believe this can strengthen the claim that ATE allows effective finetuning without large scale data.

Minor questions and remarks:

1. On page 8 (line 410), there is a missing link to the appendix.
2. The paper is pretty dense in formulae and notations (especially in the preliminary and method section). It would be better to move some of them to the appendix for presentation.
3. In eq. 5, $\mathcal{D}$ is not defined (as compared to $\mathcal{D}_\text{pretrain}$. What does it stand for?
4. On line 266, “we introduce the classifier guidance” should have “the” removed.
5. In figure 3, I think there’s a mismatch between the figure’s numbering system (6w, which I presumed to be 60k) and the description. It would be better to check these typos and correct them as well in the review period.

References:

[1] Liu, B. et al., 2023. “LIBERO: Benchmarking Knowledge Transfer for Lifelong Robot Learning”. NeurIPS

[2] Wagenmaker, A. et al., 2025. “Steering Your Diffusion Policy with Latent Space Reinforcement Learning”. CoRL.

[3] Du, M. et al., 2025. “DynaGuide: Steering Diffusion Polices with Active Dynamic Guidance”. ArXiv.

---

> ### Author Response · Authors · 2025-11-25
> **Response to Reviewer XTcT (Part 1)**
>
> Dear Reviewer XTcT,
>
> We sincerely appreciate your precious time and constructive comments. In the following, we would like to answer your concerns separately.
>
> **Strength & Discussion 1**: I believe that this type of alignment procedure in the finetuning stage is also doing something similar to that of RL. I believe that you can follow the same line of thinking proposed in CFGRL [1] and show that this form of classifier guidance in finetuning is a form of policy improvement.
>
> **Response**: We sincerely thank you for highlighting this insightful connection to CFGRL[1]. We fully agree that both CFGRL and our latent-guidance strategy share a common perspective — **improving policy behavior by reshaping the action distribution via score-based guidance**.
>
> That said, we would like to clarify an important conceptual distinction:
>
> - **CFGRL mainly performs reward-based policy improvement**. The classifier signal is derived from an advantage function $A(s,a)$, and the guided policy provably improves expected returns.
>
> - **ATE performs reward-free domain alignment**. Our guidance signal is defined solely from target-domain action demonstrations in the unified latent space and does not assume access to rewards or value functions.
>
> But interestingly, Section 6 of CFGRL (*CFGRL unlocks hidden gains in goal-conditioned behavioral cloning*) introduces a goal-conditioned behavioral cloning setting where the classifier term can be interpreted as a proxy for advantage without requiring explicit reward labels.
> We posit that under idealized assumptions, the distance metric over the unified latent space used in ATE could similarly be viewed as an equivalent proxy for advantage function.
>
> With this assumption, ATE can also be rigorously framed as performing one-step policy improvement towards a "target domain optimality", even without explicit reward labels. Instead of conditioning on a specific goal $g$, we condition on the **Target Domain Manifold** (encapsulating the target embodiment and task).
> - In CFGRL: The optimality variable $o$ implies reaching the goal $g$.
> - In ATE: The optimality variable $o$ implies aligning with the adaptation data distribution in the unified latent space.
>
> Let the "target domain optimality" $p(o|s,a)$ be proportional to the likelihood that an action $a$ belongs to the target domain optimal action distribution. The assumption suggests this likelihood is expressed by the adaptation-specific VAE encoder $E_\psi$. Assuming RBF kernel parameterization for this likelihood, the probability of an action $a$ matching a target demonstration $a^*$ is:
>
> $$p(o|s,a) \propto \exp\left( - \frac{\|E_\psi(a) - E_\psi(a^*)\|^2}{2\sigma^2} \right)$$
>
> i.e., $f(A(s,a)) \approx \exp\left( -\|E_\psi(a) - E_\psi(a^*)\|^2 \right)$, then the guidance term derived in CFGRL (Eq. 9 in [1]) becomes:
>
> $$\nabla_a \log p(o|s,a) \propto -\nabla_a \|E_\psi(a) - E_\psi(a^*)\|^2$$
>
> In order to apply the guidance from such a optimality indicator, we have to use it during the generation process of score-based model generation process. In other words, $a^k$ would be the noisy action at arbitrary diffusion timesteps $k$, and $a^*$ would thus be the action labels in the demonstration trajectory from the target domain.
> Since our setup is actually goal-conditioned behavioral cloning setup, we always take the label from the demonstration as the optimal one by default.
>
> **Then, $\nabla_a \log p(o|s,a)$ is exactly the guidance term $g$ used in ATE (Equation 11 in our paper).** Under such assumption, ATE is not merely performing domain adaptation; it is mathematically equivalent to performing policy improvement. The derivation above represents our intuitive analysis inspired by your comment. We acknowledge that there may be theoretical nuances we have not fully captured, and we would greatly value any further insights or corrections you could share to deepen our understanding of this connection.
>
> We have revised the manuscript to incorporate this theoretical connection (see Appendix A.2), citing CFGRL to strengthen the motivation of our guidance mechanism.
>
> **Reference**
>
> [1] Frans, K. et al., 2025. “Diffusion Guidance Is a Controllable Policy Improvement Operator”. ArXiv.

---

> > ### Author Response · Authors · 2025-11-25
> > **Response to Reviewer XTcT (Part 2)**
> >
> > **Weakness 1**: While the paper shows considerable improvement in aggregate, each task is only evaluated with 5 trials (correct me if I’m wrong). If possible, it is better to use a larger number of trials per task (10-20) so that there is less variance in your experiment procedure.
> >
> > **Response**: Thank you for your valuable suggestion regarding the number of evaluation trials. We would like to clarify that most of our experiments already used a sufficiently large number of trials to ensure statistical reliability. Based on your suggestion, we also conducted some additional evaluations to further validate our results w.r.t. generalization:
> >
> > - **RoboTwin 1.0 (Simulation):** 100 evaluation trials per task.
> > - **ManiSkill3 (Simulation):** 10 random seeds × 25 trials each, with the final results averaged over all seeds.
> > - **Real-World Experiments (Real-World):** 20 trials per task per checkpoint (see Appendix D.3).
> > - **Generalization Experiments:**
> >   - **Spatial Generalization:** 5 distinct spatial positions × 5 trials each, totaling 25 evaluations; the reported result is the average success rate across the 5 positions.
> >   - **Varying Illumination:** 3 illumination conditions (low, high, flickering) × 5 trials each, totaling 15 evaluations.
> > - **Additional Experiments:** For *Visual Distractor* and *Human Disturbance* conditions, we conducted 15 additional trials for each.
> >
> > The updated results are reported in Table 8. We hope this addresses your concern.
> >
> > **Weakness 2**: The method section is pretty dense in notation. I believe that the paper can be reformatted for better readability.
> >
> > **Response**: We appreciate the reviewer for this helpful suggestion. As our method spans both diffusion and flow-matching formulations, the notation in the Methodology section is intended to ensure mathematical clarity and correctness, especially when describing how ATE guidance is integrated into both formulations in a unified way.
> >
> > To improve readability while maintaining mathematical rigor, we plan to streamline certain expressions and add brief clarifying remarks where appropriate in the next revision. Additionally, if the paper is accepted, we will further refine the notation presentation in the camera-ready version (e.g., grouping shared symbols and moving secondary derivations to the appendix), ensuring improved clarity without introducing unnecessary structural changes during the rebuttal stage.
> >
> > We hope these adjustments address your concerns on clarity without compromising completeness.
> >
> > **Question 1**: How does ATE compare to other guidance methods such as DSRL or DynaGuide (although their formulations are different)?
> >
> > **Response**: We thank the reviewer for pointing out these relevant works. While DSRL [2] and DynaGuide [3] also utilize guidance mechanisms for diffusion policies, Align-Then-stEer (ATE) is fundamentally distinct in its **objective (cross-embodiment adaptation)** and **steering mechanism (unified latent alignment)**.
> >
> > We clarify the key differences below:
> >
> > 1. **Objective: Cross-Embodiment Adaptation vs. Reward/Constraint Optimization**
> > - ATE (Ours): Our primary goal is adaptation to new domains with potentially different action spaces (e.g., DOFs). ATE explicitly bridges the distribution gap between a pre-trained policy (e.g., single-arm) and a target embodiment (e.g., dual-arm) by constructing a unified latent space via asymmetric VAEs. Note that our guidance is used during training time.
> > - DSRL: Focuses on **Reinforcement Learning**. But DSRL is essentially a **Test-Time Scaling** approach which aims to select the best input noise to generate the best action mode but solves this problem in a RL manner. It treats the pre-trained diffusion policy as a fixed transformation and optimizes the input noise distribution ($w$) to maximize a reward signal. It does not inherently address the mapping between disparate action spaces (e.g., 7-DoF to 14-DoF).
> > - DynaGuide: Focuses on **Test-Time Steering** for constraints. It uses an external dynamics model to predict future states and steers the policy to satisfy specific test-time objectives (e.g., "avoid the drawer").
> >
> > 2. **Mechanism: Parameter Fine-tuning vs. Frozen Base Policy**
> > - ATE (Ours): We **fine-tune** the VLA parameters. We use classifier guidance during the training phase to steer the policy's weights toward the target distribution defined by the aligned latent space. This permanently adapts the model to the new embodiment.
> > - DSRL & DynaGuide: Both methods typically keep the base diffusion policy **frozen**. DSRL learns a separate "latent-noise policy" ( $\pi^w$ ), and DynaGuide applies gradient guidance during inference without updating the policy weights. While effective for their respective goals, they do not fundamentally transform the policy's internal representations to accommodate a new embodiment's kinematics.

---

> > > ### Author Response · Authors · 2025-11-25
> > > **Response to Reviewer XTcT (Part 3)**
> > >
> > > 3. **Handling Action Distribution Mismatch**
> > > - ATE (Ours): ATE is specifically designed to handle the "Action Distribution Shift". By utilizing the mode-seeking property of the reverse KL divergence in our VAEs (Stage 1), we force the target embodiment's actions to embed into the pre-training manifold. This alignment is missing in DSRL and DynaGuide, which operate directly on the existing action/noise space of the base model.
> > >
> > > That said, we conducted additional evaluations comparing ATE against both methods:
> > >
> > > - **Comparison with the offline version of DSRL**: We evaluated the methods on the **LIBERO-10 task suite (10 tasks, 10 trials each)**. To implement the offline version of DSRL, we used the adaptation dataset as a static buffer with sparse rewards ( $r=1$  for the final frame) to train the latent-noise critic. Without online interaction to correct value estimation errors, RL-based objectives can be sensitive and unstable when learning on limited expert data.
> > >
> > > |Benchmark| $\pi_0$ fine-tuned directly | $\pi_0$ fine-tuned directly + Offline DSRL | $\pi_0$ fine-tuned w/ ATE (Ours) |
> > > |-|-|-|-|
> > > |LIBERO 10 | 0.78 | 0.43 | **0.88** |
> > >
> > > - **Comparison with DynaGuide**: We compared Diffusion Policy (DP) baselines against DynaGuide and ATE on three specific tasks. For a fair comparison, we modified DynaGuide to support multi-view inputs and tuned its hyperparameters ($\alpha$, scale) for best performance. While DynaGuide improves performance on some tasks, it is unstable on others (e.g., degrading dual bottles pick from $0.32 \rightarrow 0.18$). ATE achieves consistent improvements across all tasks.
> > >
> > >
> > > |           Task           | DP | + DynaGuide | + ATE (ours) |
> > > |--------------------------|-------------|-------------|--------------|
> > > | put_apple_cabinet        |     0.29    |    0.46     |   **0.46**   |
> > > | empty_cup_place          |     0.22    |    0.30     |   **0.37**   |
> > > | dual bottles pick(easy)  |     0.32    |    0.18     |   **0.43**   |
> > >
> > > **Question 2**: Can authors also discuss how the sample efficiency of this method stack up compared to the baseline methods across various finetuning set sizes? I believe this can strengthen the claim that ATE allows effective finetuning without large scale data.
> > >
> > > **Response**: Thanks for the insightful question.
> > > We evaluated the success rates of the $\pi_0$ backbone with and without ATE on the RoboTwin benchmark using varying dataset sizes (10, 25, and 50 episodes per task).
> > >
> > > |                        | 10 Episodes / Task |      | 25 Episodes / Task |      | 50 Episodes / Task |      |
> > > |------------------------|--------------------|------|--------------------|------|--------------------|------|
> > > | Task                   | $\pi_0$                 | $\pi_0$ + ATE | $\pi_0$                 | $\pi_0$ + ATE | $\pi_0$                 | $\pi_0$ + ATE |
> > > | Put Apple Cabinet      |                  0 |    0 |               0.02 | **0.27** |               0.34 | **0.55** |
> > > | Blocks Stack Easy      |                  0 | **0.01** |               0.05 | **0.06** |                0.3 |  **0.5** |
> > > | Diverse Bottles Pick   |               0.07 | 0.07 |               0.04 | **0.22** |                0.2 |  **0.4** |
> > > | Dual Bottles Pick Easy |               0.05 | **0.14** |               0.21 | **0.59** |               0.48 | **0.85** |
> > > | Block Handover         |               0.01 | 0.01 |               0.14 | **0.31** |                0.8 | **0.92** |
> > >
> > > Key Findings:
> > >
> > > 1. **Significant Improvement in Low-Data Regimes**: With only 25 episodes, the baseline fails on complex tasks like Put Apple Cabinet (2%), whereas ATE achieves a 27% success rate.
> > > 2. **2x Sample Efficiency**: ATE often matches or exceeds the baseline's "50-episode performance" using only 25 episodes (e.g., Dual Bottles Pick Easy: 0.59 vs. 0.48). This demonstrates that ATE effectively accelerates convergence when data is scarce.

---

> > > > ### Author Response · Authors · 2025-11-25
> > > > **Response to Reviewer XTcT (Part 4)**
> > > >
> > > > **Minor Questions**: (i) On page 8 (line 410), there is a missing link to the appendix. (ii) The paper is pretty dense in formulae and notations (especially in the preliminary and method section). It would be better to move some of them to the appendix for presentation. (iii) In eq. 5, $\mathcal{D}$ is not defined (as compared to $\mathcal{D}_{\rm pretrain}$. What does it stand for? (iv) On line 266, “we introduce the classifier guidance” should have “the” removed. (v) In figure 3, I think there’s a mismatch between the figure’s numbering system (6w, which I presumed to be 60k) and the description. It would be better to check these typos and correct them as well in the review period.
> > > >
> > > > **Response**: Thank you for the detailed suggestions. We address each point below:
> > > >
> > > > (i) We have corrected the missing Appendix reference at line 410 and will highlight this change in the revised manuscript.
> > > >
> > > > (ii) Thank you for pointing out the density of notation. While the current formulation ensures methodological completeness, we acknowledge that reducing the burden of mathematical symbols can further improve accessibility. We plan to reorganize several expressions and add clarifications in the next revision. Moreover, if the paper is accepted, we will refine notation in the camera-ready version—e.g., grouping shared symbols and moving secondary derivations to the appendix — while avoiding an excessive amount of modifications during the review period.
> > > >
> > > > (iii) We apologize for the confusion. In Eq. 5, $\mathcal{D}$ denotes a dataset from which the expectation is taken. On the left-hand side of Eq. 5, we explicitly specify that $\mathcal{D}$ is instantiated as $\mathcal{D}_{\rm adaptation}$, referring to “a narrow but high-quality dataset collected from the target embodiment for downstream applications,” as defined in Section 3 (*Training objective of VLAs.*).
> > > >
> > > > (iv) Thank the reviewer for catching the typo — we have removed “the” in that phrase.
> > > >
> > > > (v) We appreciate the reviewer pointing out the mismatch between “6w” (short for 60k) and the figure caption. We have corrected the numbering and reformat the figure axis labels to avoid ambiguity.
> > > >
> > > > All modifications are applied in the revised version and highlighted in **red** for easy verification.

---

> ### Comment · Reviewer_XTcT · 2025-11-27
>
> Thank you for your detailed feedback, which have addressed most of my concerns. I have decided to raise the score (I'm not sure how to change it on the portal, but the rating will now be 8).

---

> > ### Author Response · Authors · 2025-11-28
> > **Sincere thanks!**
> >
> > We are truly grateful for your support and for your decision to raise the score. We are glad that our responses have addressed most of your concerns.
> >
> > It is unfortunate that the rating-editing entrance is now frozen due to the terrible reviewer profile hacking issue (we learned about it from the SNS). Nevertheless, we sincerely believe that **your written confirmation here (rating = 8) can be sufficient for the Area Chair** and greatly helps ensure a fair assessment of our work.
> >
> > If you have any remaining concerns or would like to discuss any aspect further, please feel free to reach out—we really appreciate the opportunity to discuss with you. Your feedback has offered us valuable insights and has significantly enriched our paper.
> >
> > Thank you again for your thoughtful engagement.
> >
> > Best regards,
> >
> > The authors

---

### Official Review · Reviewer_tPwB · 2025-11-02

**Soundness:** 3
**Presentation:** 2
**Contribution:** 2
**Rating:** 4
**Confidence:** 2

**Summary:**

This paper proposes two-step framework, namely ATE, on efficient robot learning on robot manipulation tasks. ATE consists of two sub-modules: (i) a dual-VAE module for robot action chunk pre-training and downstream action chunk adaptation, where the embedding space of the latter one is trained to align with the pre-trained VAE. (ii) a steering module for diffusion- and flow-based VLA models. ATE is a plug-and-play framework and could be directly applied to different VLA models. Simulation and real-world experiments demonstrate consistent performance gain w.r.t diffusion- and flow-based models.

**Strengths:**

1. The proposed method, including the dual VAE and steer module is theoretically sound.
2. The size of ATE is relatively small w.r.t base models like pi_0, yet it brings consistent performance gain over the base models.
3. Extensive experiments make ATE technically solid.

**Weaknesses:**

I found this paper difficult to follow. Please refer to the questions.

**Questions:**

Questions:
1.	It seems that there lacks necessary (and straightforward) illustration of how diffusion- and flow-based methods like pi_0 work, and how ATE is trained/integrated with these models.
2.	What’s the necessity of training two VAEs? Once the first VAE is pre-trained using large-scale datasets like OpenXE, it could theoretically transfer and fit-in to a smaller downstream dataset. The motivation could be more clearly illustrated (even though it is mentioned at line 254).
3.	Is it possible to directly apply the steering module to other latent-based methods (like Q.2)? Or, is it strongly tied to work with the proposed InfoVAE?
Suggestions:
1.	Font size of Figure 1 and Figure 2 is too small to clearly recognize the text.
2.	Line 410: missing reference
3.	Line 457: ATE in inconsistent font.

---

> ### Author Response · Authors · 2025-11-25
> **Response to Reviewer tPwB (Part 1)**
>
> Dear Reviewer tPwB,
>
> We sincerely appreciate your precious time and constructive comments. In the following, we would like to answer your concerns separately.
>
> **Question 1**: It seems that there lacks necessary (and straightforward) illustration of how diffusion- and flow-based methods like pi_0 work, and how ATE is trained/integrated with these models.
>
> **Response**: Thanks for the questions. Below, we provide point-by-point responses:
> 1. **Mechanism of flow- or diffusion-based VLAs ($\pi_0$ and RDT)**. Both $\pi_0$ and RDT diverge models the policy as a conditional generative process $p(a_t | o_t, l)$, where $a_t$ is the action chunk, $o_t$ is the observation, and $l$ is the instruction.
> - $\pi_0$ (flow-based): $\pi_0$ learns a velocity field $v_\theta$ that defines a probability path from a standard Gaussian distribution to the conditional action chunk distribution. During inference, it generates actions by solving an Ordinary Differential Equation (ODE) (e.g., using Euler integration) starting from noise.
> - RDT (diffusion-based): RDT employs a Diffusion Transformer to learn a noise prediction network $\epsilon_\theta$. Inference involves an iterative denoising process (e.g., DDPM, DDIM or DPM-Solver) to progressively recover clean actions from Gaussian noise.
>
> 2. **Plug-and-Play Integration of ATE**. We clarify that ATE is designed as a **model-agnostic, plug-and-play module** that requires **no architectural modifications** to the score-based pre-trained VLA. Instead, ATE operates by modifying the Supervised Fine-Tuning (SFT) objective:
> - **Standard SFT**: Typically minimizes a flow-matching loss (for $\pi_0$, shown as Eq. 3) or a denoising loss (for RDT, shown as Eq. 1) on the target dataset.
> - **ATE-Enhanced SFT**: We inject a guidance term $g$ directly into the original objective. This allows us to explicitly steer the generative process toward the target distribution and enforce it at the training time.
>
> 3. **Implementation Workflow**. The training and integration of ATE follow a rigorous two-stage protocol:
> - **Stage 1: Alignment via Unified Action Latent Space**. Before fine-tuning the VLA, we train two lightweight Variational Autoencoders (VAEs) to construct a unified latent space: (i) Pre-training action VAE: Trained on the large-scale pre-training dataset to capture a diverse action latent prior. (ii) Adaptation VAE: Trained on the small-scale target dataset. Crucially, we apply a Reverse KL Divergence constraint to force the target actions' latent embeddings to align with the modes of the pre-training latent distribution. Note that This stage is computationally efficient.
> - **Stage 2: Steering via Guidance (Frozen VAEs)**. During the SFT of the VLA, we freeze the dual VAEs and use them solely to compute the guidance signal $g$. (i) Latent Projection: At each training step, the noisy action estimate $\hat{a}^k$ and the ground truth target action $a^0$ are projected into the latent space using the frozen Adaptation VAE encoder $E_\psi$. (ii) Guidance Computation: We calculate the gradient of the latent discrepancy via Eq. 11. (iii) Loss Correction: This gradient $g$ is added to the VLA's objective function (Eq. 7 for diffusion-based VLAs, Eq. 10 for flow-based VLAs), explicitly steering the policy toward the aligned target distribution.

---

> > ### Author Response · Authors · 2025-11-25
> > **Response to Reviewer tPwB (Part 2)**
> >
> > **Question 2**: What’s the necessity of training two VAEs? Once the first VAE is pre-trained using large-scale datasets like OpenXE, it could theoretically transfer and fit-in to a smaller downstream dataset. The motivation could be more clearly illustrated (even though it is mentioned at line 254).
> >
> > **Response**: Thanks for the reviewer’s insightful question regarding the necessity of the dual-VAE design. While fine-tuning a single VAE is theoretically feasible, we would like to highlight that training two distinct VAEs is essential to address the heterogeneity of action spaces and to enable mode-seeking latent alignment.
> >
> > We clarify the motivation from the following two perspectives:
> > 1. **Bridging the Physical Heterogeneity Gap**. The pre-training datasets (e.g., Open X-Embodiment) and the downstream adaptation datasets often originate from fundamentally different embodiments (e.g., single-arm vs. dual-arm robots)
> > - **Limitation of Single VAE Fine-tuning**: The pre-training VAE encoder $E_{\phi}$ is optimized to map pre-training actions to a latent manifold $Z$. If we simply fine-tune this encoder on the target domain, the drastic physical discrepancies (e.g., different DOFs) may cause the target actions to map to undefined or low-density regions of the latent space or destroy the structured latent manifold, degrading the pre-trained priors.
> > - **Necessity of the Second VAE**: Instead, the Adaptation VAE learns a mapping tailored to the specific embodiment, projecting target actions into the existing structured latent geometry without distorting it.
> >
> > 2. **Alignment via Mode-Seeking (Reverse KL Divergence)**. The core contribution of ATE is utilizing Reverse KL Divergence for distribution alignment, which requires a fixed target distribution to align against.
> > - **Mechanism**: We treat the learned latent distribution of the first VAE, $q_{\phi}(z)$, as a fixed prior. We then optimize the second VAE using the reverse KL objective: $\min D_{KL}(q_{\psi}(z|\tilde{a}) || q_{\phi}(z))$ in Eq. 5.
> > - **Necessity**: As highlighted in the paper, Reverse KL induces mode-seeking behavior. This explicitly forces the latent embeddings of the adaptation actions to cluster tightly around the valid "modes" of the pre-training distribution. A single VAE fine-tuning approach typically lacks this explicit constraint to embed new data into the valid modes of the learned prior.
> >
> > Furthermore, our ablation in Appendix D.5 empirically supports this design choice: using a single VAE leads to notably reduced adaptation performance, demonstrating that effective distribution alignment cannot be achieved without a fixed latent prior. Although the single-VAE version in our ablation is trained from scratch, the decline in downstream task success rates strongly indicates that replacing the dual-VAE design with fine-tuning a single VAE would sacrifice latent geometry and alignment quality, making cross-embodiment adaptation substantially less reliable.
> >
> >
> > **Question 3**: Is it possible to directly apply the steering module to other latent-based methods (like Q.2)? Or, is it strongly tied to work with the proposed InfoVAE?
> >
> > **Response**: We thank the reviewer for the insightful question. Our steering module is **not constrained to InfoVAE**; it is designed as a **plug-and-play guidance mechanism** that can operate on any latent space with meaningful structure.
> >
> > To verify this generality, we replaced InfoVAE with a **vanilla VAE** while keeping our alignment-learning stage and steering module unchanged. The results are summarized below:
> >
> > | Backbone | Task | w/o VAE | + Vanilla VAE | + InfoVAE (ours) |
> > |---------|------|---------|---------------|-----------------|
> > | DP | dual bottles pick easy | 0.32 | 0.39 | **0.43** |
> > | DP | shoe_place | 0.14 | 0.18 | **0.22** |
> > | DP | put_apple_cabinet | 0.29 | 0.42 | **0.46** |
> > | DP | empty_cup_place | 0.22 | 0.34 | **0.37** | 0.30|
> > | $\pi_0$ | blocks stack easy | 0.30 | 0.31 | **0.50** |
> > | $\pi_0$ | diverse bottles pick | 0.20 | 0.36 | **0.40** |
> > | $\pi_0$ | dual bottles pick easy | 0.48 | 0.81 | **0.85** |
> > | $\pi_0$ | block handover | 0.80 | **0.94** | 0.92 |
> >
> >
> > These findings confirm three important properties of our design:
> >
> > 1. **General compatibility**: The steering module consistently improves the success rate even when paired with a standard VAE, demonstrating that it *directly applies to other latent-based methods* without architectural changes.
> >
> > 2. **Why InfoVAE performs better**: Standard VAEs commonly suffer from **inaccurate amortized latent inference**, leading to poorly structured latent representations. This effect prevents reliable cross-domain latent alignment, which is essential for correct steering in the target domain. InfoVAE alleviates this problem by enforcing stronger mutual-information constraints, forming a more structural latent structure. Thus, steering becomes more stable and effective.

---

> > > ### Author Response · Authors · 2025-11-25
> > > **Response to Reviewer tPwB (Part 3)**
> > >
> > > 3. **Modularity as a design choice**: The key contribution of our approach lies in **factorizing latent space alignment and downstream steering**. Better latent models naturally bring larger gains, making our approach adoptable with stronger latent learning methods.
> > >
> > > In summary, the steering module is **general and not tied to InfoVAE**, while InfoVAE simply enables a more semantically aligned latent space that amplifies the steering benefits. We have included these results in the revised version of the paper (See **Appendix J.4** for details).
> > >
> > >
> > > **Suggestions 1-3**: Typos issues and presentation issues
> > >
> > > **Response**: We thank the reviewer for pointing out these presentation issues. We have carefully addressed all of them in the updated manuscript (changes highlighted in **red**):
> > >
> > > 1. **Figure font sizes**: We enlarged the font sizes in Figure 1 and Figure 2 to ensure text readability, especially in the online zoom-in view.
> > >
> > > 2. **Missing reference at Line 410**: The missing citation has been added and is now properly referenced.
> > >
> > > 3. **Inconsistent font for “ATE” at Line 457**: The formatting has been corrected for consistency.

---

> > > > ### Comment · Reviewer_tPwB · 2025-11-26
> > > >
> > > > I thank the authors for their detailed explaination, which addressed my concerns. Hence, I decide to raise my score to 6. I suggest that these details could be further discussed in the final manuscript.

---

> ### Author Response · Authors · 2025-11-26
> **Thank you for the positive feedback!**
>
> We sincerely thank you for your continued engagement and for raising the score to 6. We are delighted to hear that our detailed explanations—particularly regarding the integration with $\pi_0$ / RDT, the necessity of the dual-VAE design, and the generality of the steering module—have successfully addressed your concerns.
>
> We fully agree with your suggestion to incorporate these details into the final manuscript. And we have integrated the vanilla VAE ablation results (demonstrating the plug-and-play nature), and the justifications for the dual-VAE architecture into the revised manuscript to ensure the paper is rigorous. See our newly added appendix J for details.
>
> Since we have fully resolved the initial concerns regarding clarity and motivation, and considering ATE’s demonstrated strengths—**plug-and-play compatibility, incurring no additional inference latency, and significant real-world performance gains (+32%)** in challenging cross-embodiment settings—we believe this work represents a substantial contribution to efficient VLA adaptation. We hope these factors might encourage a further re-evaluation towards an 'Accept (8)'.
>
> Please feel free to reach out if any further questions arise. And thank you again for your valuable time and insights.
>
> Sincerely,
>
> The Authors

---

### Official Review · Reviewer_Udtp · 2025-11-06

**Soundness:** 3
**Presentation:** 3
**Contribution:** 2
**Rating:** 6
**Confidence:** 3

**Summary:**

The paper proposes ATE, a plug-and-play adaptation framework for vision-language-action (VLA) models. Stage 1 “Align” learns a unified action latent space by training an InfoVAE on pre-training actions and a second InfoVAE on target-domain actions with a reverse-KL regularizer toward the pre-training latent distribution. Stage 2 “Steer” adds a training-time guidance term that pushes generated action chunks toward target-domain actions by penalizing the squared distance between their latent encodings. The method claims minimal architectural changes and low overhead.
The method claims minimal architectural changes and low overhead. It is evaluated on RoboTwin 1.0 (17 tasks), ManiSkill3 (2 tasks), and a real dual-arm RealMan platform (4 tasks + 1 tool-use).
It achieves 9.8% and 32% success rate improvement on Simulation and real-world setting, respectively. There is also a qualitative claim of smoother, safer motions (force plots).

**Strengths:**

- Tackles an important and practical problem: efficient adaptation of pretrained VLAs across embodiments and tasks.
- Simple, model-agnostic recipe that integrates with both diffusion and flow matching, with no architectural changes.
- Clear implementation description and algorithm boxes; the method is plausibly easy to reproduce and deploy.
- Writing is good and the paper is easy to follow.
- The improvements look good, and it seems like a good RL alternative.

**Weaknesses:**

- I feel like the proposed method is more like a self-distillation method which shares similar objective with RL. I wonder whether it can compete with existing offline RL methods. Now the model did not show any other baseline methods besides the model it based on so it is hard to tell.

**Questions:**

Misc:
Missing Appendix ref in L410

---

> ### Author Response · Authors · 2025-11-25
> **Response to Reviewer Udtp (Part 1)**
>
> Dear Reviewer Udtp,
>
> We sincerely appreciate your precious time and constructive comments. In the following, we would like to answer your concerns separately.
>
> **Weakness 1**: I feel like the proposed method is more like a self-distillation method which shares similar objective with RL. I wonder whether it can compete with existing offline RL methods. Now the model did not show any other baseline methods besides the model it based on so it is hard to tell.
>
>
> **Response**: Thanks for the comment. We are pleased to discuss the differences and connections between our approach and RL. **(1) Difference to self-distillation methods.**  In RL, self-distillation typically involves using the agent’s learned policy to rollout trajectories in the environment and collect diverse experience [1]. The policy is then refined by either imitating high-return trajectories or by substituting auxiliary goals in a goal-conditioned RL framework [2]. In contrast, our objective is to adapt a pre-trained VLA policy to a adaptation domain whose underlying policy potentially derived from human demonstrations, scripted controllers, or other sources, which can differ significantly from the pre-trained VLA. This discrepancy arises naturally in our cross-embodiment and cross-task settings. Moreover, self-imitation methods are inherently limited by the capabilities of the rollout policy, while ATE goes beyond this constraint by performing explicit adaptation in the target domain and leverages SFT with domain-specific guidance, enabling the final policy to diverge meaningfully from the pre-trained behaviors. **(2) Comparison to offline RL.** Offline RL offers a powerful paradigm for learning policies from fixed datasets without environment interaction. However, it is ill-suited to our cross-domain adaptation setting. Our problem is not merely about learning from a static dataset, but about bridging a substantial distributional gap between the pre-training and target domains—particularly under cross-embodiment conditions. While recent advances in cross-domain offline RL propose to address such distribution shifts by modeling value functions or environment dynamics to quantify domain discrepancies, these approaches often entail complex auxiliary learning objectives [3,4]. In contrast, our method sidesteps these requirements: it only necessitates aligning the source and target domains into a unified action space and applying a lightweight guidance mechanism to steer the pre-trained policy toward target-domain behaviors.
>
> To empirically validate this, we added an experiment comparing ATE with DSRL [5], a state-of-the-art RL method designed for steering diffusion policies, and DynaGuide [6], which also introduces a classifier guidance steering mechanism but uses it during inference time.
>
> - **Comparison with the offline version of DSRL**: We evaluated the methods on the **LIBERO-10 task suite (10 tasks, 10 trials each)**. To implement the offline version of DSRL, we used the adaptation dataset as a static buffer with sparse rewards ($r=1$ for the final frame) to train the latent-noise critic. Without online interaction to correct value estimation errors, RL-based objectives can be sensitive and unstable when learning on limited expert data.
>
> |Benchmark| $\pi_0$ fine-tuned directly | $\pi_0$ fine-tuned directly + Offline DSRL | $\pi_0$ fine-tuned w/ ATE (Ours) |
> |-|-|-|-|
> |LIBERO 10 | 0.78 | 0.43 | **0.88** |
>
> - **Comparison with DynaGuide**: We compared Diffusion Policy (DP) baselines against DynaGuide and ATE on three specific tasks. For a fair comparison, we modified DynaGuide to support multi-view inputs and tuned its hyperparameters ($\alpha$, scale) for best performance. While DynaGuide improves performance on some tasks, it is unstable on others (e.g., degrading dual bottles pick from $0.32 \rightarrow 0.18$). ATE achieves consistent improvements across all tasks.
>
>
> |           Task           | DP | + DynaGuide | + ATE (ours) |
> |--------------------------|-------------|-------------|--------------|
> | put_apple_cabinet        |     0.29    |    0.46     |   **0.46**   |
> | empty_cup_place          |     0.22    |    0.30     |   **0.37**   |
> | dual bottles pick(easy)  |     0.32    |    0.18     |   **0.43**   |

---

> > ### Author Response · Authors · 2025-11-25
> > **Response to Reviewer Udtp (Part 2)**
> >
> > **Question 1**: Misc: Missing Appendix ref in L410
> >
> > **Response**: Thanks for pointing this out. We have corrected the missing Appendix reference in the revised manuscript, and the change is highlighted in **red** for easy verification.
> >
> > **Reference**
> >
> > [1] Oh, Junhyuk, et al. "Self-imitation learning." International conference on machine learning. PMLR, 2018.
> >
> > [2] Liu, Minghuan, Menghui Zhu, and Weinan Zhang. "Goal-conditioned reinforcement learning: Problems and solutions." arXiv preprint arXiv:2201.08299 (2022).
> >
> > [3] Xu K, Bai C, Ma X, et al. Cross-domain policy adaptation via value-guided data filtering. NeurIPS 2023
> >
> > [4] Wen X, Bai C, Xu K, et al. Contrastive representation for data filtering in cross-domain offline reinforcement learning. ICML 2024
> >
> > [5] Wagenmaker, Andrew, et al. "Steering Your Diffusion Policy with Latent Space Reinforcement Learning." arXiv preprint arXiv:2506.15799 (2025).
> >
> > [6] Du, Maximilian, and Shuran Song. "DynaGuide: Steering Diffusion Polices with Active Dynamic Guidance." arXiv preprint arXiv:2506.13922 (2025).

---

### Author Response · Authors · 2025-11-25
**Extensive New Experiments and Analysis & Sry for the delay in our response**

Dear Reviewers,

We sincerely apologize for the delay in our response. We have spent the past weeks conducting extensive additional experiments to thoroughly address the constructive feedback regarding baselines, sample efficiency, and ablation studies. We believe these new results significantly strengthen the paper's claims.

Below is a summary of the new experiments and analyses included in our individual responses and the revised manuscript:

1. **Comparison with State-of-the-Art Guidance Methods:**
- **vs. DSRL (Offline RL)**: We evaluated ATE against the offline version of DSRL on the LIBERO-10 benchmark. Results show ATE significantly outperforms DSRL ($0.88$ vs. $0.43$ success rate), demonstrating the superiority of distribution alignment over reward-based offline RL in positive-only demonstration regimes.
- **vs. DynaGuide (Inference Steering)**: We compared ATE with DynaGuide on multi-view tasks. ATE achieves higher and more stable success rates while **incurring zero inference latency**, whereas DynaGuide severely impacts real-time control speed.

2. **Sample Efficiency Analysis**:
- We conducted a systematic evaluation on the RoboTwin benchmark with varying dataset sizes **(10, 25, and 50 episodes per task)**.
- **Key Finding**: ATE achieves **2x sample efficiency**, matching the baseline's 50-episode performance using only 25 episodes, and preventing policy collapse in extreme low-data regimes.

3. **Robustness Under Large Kinematic Gaps**:
- We added a "toy setup" transferring a policy pre-trained on **Dual-Arm** data (RoboTwin) to a **Single-Arm** task (ManiSkill). ATE achieved an **82%** success rate (+10% over direct fine-tuning), proving the alignment's effectiveness even under significant embodiment shifts.

4. **Ablation Studies on VAE Design**:
- **Vanilla VAE vs. InfoVAE**: We verified that our InfoVAE-based design significantly outperforms Vanilla VAE, confirming the necessity of structured latent alignment.
- **Hyperparameter Sensitivity**: We provided sweeps for the reverse KL penalty weight ($\alpha$) and guidance scale ($\lambda$), identifying robust operating ranges.

5. **Latent Space Visualization**:
- We added **t-SNE visualizations** (Appendix J.4) illustrating how InfoVAE effectively aligns the adaptation distribution with the pre-training modes compared to Vanilla VAE.


We have updated the manuscript (changes highlighted in red) to incorporate these results. We thank the reviewers again for their insightful comments which have greatly improved the quality of our work.

Sincerely,

The Authors

---

### Author Response · Authors · 2025-12-02
**Summary of Rebuttal Updates**

Dear PCs, SACs, ACs and Reviewers,

We would like to express our sincere appreciation for the reviewers' time and thoughtful feedback. As the author-reviewer discussion phase comes to a close, we would like to provide a general summary of the reviews and outline the efforts we have made during the discussion phase.

Upon the release of the initial reviews, we are encouraged that the reviewers find our framework and analysis to be **"theoretically sound"** (Reviewer tPwB), **"well-grounded"** (Reviewer RFTf), and tackling an **"important and practical problem"** (Reviewer Udtp). They commend the proposed ATE method as a **"simple, model-agnostic recipe"** (Reviewer Udtp) that brings **"consistent performance gain"** (Reviewer tPwB) with **"extensive experiments"** (Reviewer XTcT) validating its robustness.

Then, we have dedicated the past few weeks to conducting extensive additional experiments and analyses to thoroughly address the constructive feedback from all reviewers. We believe these new results significantly strengthen the empirical rigor and theoretical clarity of our paper.

We summarize the key updates and additional experiments included in our revised manuscript and individual responses below:

| Category | Reviewers' Concerns | Our Additional Experiments & Responses |
| :--- | :--- | :--- |
| **Baselines & Comparisons** | **Reviewer Udtp & RFTf:** Lack of comparison with other guidance-based methods (e.g., DSRL, DynaGuide) and offline RL baselines. | **1. vs. Offline DSRL:** We evaluated ATE against the offline version of DSRL on the **LIBERO-10** benchmark. ATE significantly outperforms DSRL (**0.88 vs. 0.43** success rate), demonstrating the superiority of our distribution alignment approach over reward-based offline RL. **2. vs. DynaGuide:** We compared ATE with DynaGuide on multi-view manipulation tasks. ATE achieves higher and more stable success rates while incurring **zero inference latency**, whereas DynaGuide severely impacts real-time control speed due to test-time gradient calculation. |
| **Sample Efficiency** | **Reviewer Udtp:** Request for analysis on sample efficiency across various fine-tuning set sizes. | **Sample Efficiency Analysis:** We conducted a systematic evaluation on the **RoboTwin** benchmark with varying dataset sizes (**10, 25, and 50 episodes per task**). **Key Finding:** ATE demonstrates **~2x sample efficiency**, matching the baseline's 50-episode performance using only **25 episodes**, and effectively prevents policy collapse in extreme low-data regimes. |
| **Robustness & Generalization** | **Reviewer RFTf:** Concerns about robustness under large kinematic/semantic gaps between pre-training and fine-tuning data. | **Dual-Arm $\to$ Single-Arm Transfer:** We added a "toy setup" transferring a policy pre-trained on **Dual-Arm** data (RoboTwin. Duarl-arm Agilex) to a **Single-Arm** task (ManiSkill Franka). Despite the significant structural mismatch, ATE achieved an **82%** success rate (**+10%** over direct fine-tuning), validating the robustness of our alignment mechanism even under drastic embodiment shifts. |
| **Methodological Ablations** | **Reviewer tPwB:** Necessity of the dual-VAE design and the choice of InfoVAE.  **Reviewer RFTf:** Effect of the reverse KL constraint. | **1. VAE Choice:** We conducted an ablation study replacing InfoVAE with Vanilla VAE. Results confirm that **InfoVAE** produces a more structured latent space (visualized via **t-SNE** in Appendix M), which is critical for effective cross-domain alignment. **2. Hyperparameter Sensitivity:** We provided comprehensive sweeps for the reverse KL penalty weight ($\alpha$) and guidance scale ($\lambda$), identifying robust operating ranges for stable adaptation. |
| **Theoretical Connection** | **Reviewer XTcT:** Connection to CFGRL and policy improvement. | **Theoretical Discussion:** We have incorporated a discussion (Appendix A.2) linking our latent guidance to the advantage-based guidance in CFGRL, intuitively framing ATE as a form of reward-free policy improvement towards the "target domain optimality". |
| **Evaluation Reliability** | **Reviewer XTcT:** Suggestion to increase the number of evaluation trials. | **Expanded Evaluation:** We significantly increased the number of trials for our experiments: - **RoboTwin:** 100 trials/task. - **ManiSkill:** 10 seeds $\times$ 25 trials. - **Real-World:** 20 trials/task. - **Generalization:** 25 trials for spatial, 15 for illumination/distractors. |

We have revised the manuscript to incorporate these new results and clarifications (changes highlighted in **red**). We would like to highlight that **Reviewer XTcT has raised their score to 8**, and **Reviewer tPwB has raised their score to 6** following our responses. We are deeply grateful for the reviewers' detailed and insightful comments, which have driven us to significantly improve the quality and completeness of our work.

Best regards,

The Authors

---

### Meta-Review · Area_Chair_mHQE · 2026-01-07

**Summary:**

Reviewers uniformly acknowledged the paper’s relevance in addressing efficient adaptation of embodied vision language action models, praising its simplicity, plug-and-play design, solid theoretical motivation, and strong empirical results across both simulation and real world settings. In the end, all four reviewers provided positive feedback.

**Reviewer Concerns:**

The authors comprehensively addressed all weaknesses and questions raised by the reviewers in their rebuttal.

**Reviewer Scores:**

Reviewer Udtp (initial 6) may raise their score by 1 point given the new baseline comparisons.
Reviewer tPwB (initial 4) has already increased their score to 6 after the rebuttal and is unlikely to adjust further.
Reviewer XTcT (initial 6) confirmed they will raise their score by 2 points to 8.
Reviewer RFTf (initial 6) may raise their score by 1 or 2 points following the additional experiments and clarifications.

---

### Decision · Program_Chairs · 2026-01-26

Accept (Poster)